# Tracing the origin of the oxygen-consuming organic matter in the hypoxic zone in a large eutrophic estuary: the lower reach of the Pearl River Estuary, China

Jianzhong Su[1], Minhan Dai[1*], Biyan He[1,2], Lifang Wang[1], Jianping Gan[3], Xianghui Guo[1], Huade Zhao[1] and Fengling Yu[1]

[1]State Key Laboratory of Marine Environmental Science, Xiamen University, Xiamen, China
[2]College of Food and Biological Engineering, Jimei University, Xiamen, China
[3]Department of Mathematics and Division of Environment, Hong Kong University of Science and Technology, Kowloon, Hong Kong, China

*Correspondence to*: Minhan Dai (mdai@xmu.edu.cn)

**Abstract.** We assess the relative contributions of different sources of organic matter, marine vs. terrestrial, to oxygen consumption in an emerging hypoxic zone in the lower Pearl River Estuary (PRE), a large eutrophic estuary located in Southern China. Our cruise, conducted in July 2014, consisted of two legs before and after the passing of Typhoon Rammasun, which completely de-stratified the water column. The stratification recovered rapidly, within one day after the typhoon. We observed algal blooms in the upper layer of the water column and hypoxia underneath in bottom water during both legs. Repeat sampling at the initial hypoxic station showed severe oxygen depletion down to 30 $\mu$mol kg$^{-1}$ before the typhoon and a clear drawdown of dissolved oxygen after the typhoon. Based on a three end-member mixing model and the mass balance of dissolved inorganic carbon and its isotopic composition, the $\delta^{13}$C of organic carbon remineralized in the hypoxic zone was -23.2 ±1.1 ‰. We estimated that 65 ±16 % of the oxygen-consuming organic matter was derived from marine sources, and the rest (35 ±16 %) was derived from the continent. In contrast to a recently studied hypoxic zone in the East China Sea off the Changjiang Estuary where marine organic matter stimulated by eutrophication dominated oxygen consumption, here terrestrial organic matter significantly contributed to the formation and maintenance of hypoxia. How varying amounts of these organic matter sources drive oxygen consumption has important implications for better understanding hypoxia and its mitigation in bottom waters.

# 1 Introduction

The occurrence of hypoxia has been exacerbated worldwide (Nixon, 1995; Diaz and Rosenberg, 2008; Rabalais et al., 2010; Zhang et al., 2013). In recent decades, more than 400 coastal hypoxic systems have been reported with an exponential growth rate of $5.5 \pm 0.23$ % $yr^{-1}$, demonstrating their persistence and complexity with respect to both science and management (Diaz and Rosenberg, 2008; Vaquer-Sunyer and Duarte, 2008). Hypoxia may not only reduce biodiversity and endanger aquatic and benthic habitats, but also alter the redox chemistry in both the water column and the underlying sediments, triggering the release of secondary pollutants (Breitburg, 2002; Rutger et al., 2002). Moreover, the management and recovery of these systems is complicated due to the hysteresis of hypoxic conditions, and the varying timescales of biological loss (within hours to weeks) and recovery from hypoxia (from months to years) (Steckbauer et al., 2011).

Coastal hypoxia usually occurs in stratified water columns where the downward mixing of oxygen from the surface is impeded (Kemp et al., 2009). Below the pycnocline, aerobic respiration is usually the predominant sink of oxygen. Organic matter, which consumes dissolved oxygen (DO) as it becomes oxidized, is thus the ultimate cause of hypoxia under favourable physical settings (Rabouille et al., 2008; Rabalais et al., 2014; Qian et al., 2016). The organic carbon (OC) that fuels respiration-driven reduction of oxygen in these systems could originate from either eutrophication-induced primary production (marine OC; $OC_{mar}$), or naturally and/or anthropogenically driven delivery from terrestrial environments (terrestrial OC; $OC_{terr}$) (Paerl, 2006; Rabalais et al., 2010).

The question of how much OC in hypoxic zones is supplied from on-site primary production versus the quantity derived from terrestrial sources has been an issue of debate (Wang et al., 2016). Much of the phytoplankton-centric hypoxia literature suggests that $OC_{mar}$ dominates oxygen consumption in hypoxic zones, owing to its higher microbial availability than $OC_{terr}$ (Zimmerman and Canuel, 2000; Boesch et al., 2009; Carstensen et al., 2014). Wang et al. (2016) quantified for the first time the relative contributions of particulate $OC_{mar}$ ($POC_{mar}$) and particulate $OC_{terr}$ ($POC_{terr}$) in consuming DO in the bottom waters of the East China Sea (ECS) off the Changjiang Estuary (CJE), and found that $POC_{mar}$ dominated DO consumption. However, other studies suggest that $POC_{terr}$ may also play an important role (Swarzenski et al., 2008;

Bianchi, 2011a; Bianchi et al., 2011b). It is thus very important to quantify the relative contributions of organic matter ($OC_{mar}$ vs. $OC_{terr}$) driving the onset and maintenance of hypoxia in coastal systems, since reducing organic matter vs. nutrient inputs requires a different set of management strategies.

The Pearl River Estuary (PRE, 21.2 °N–23.1 °N, 113.0 °E–114.5 °E) is surrounded by several large cities including Hong Kong, Shenzhen and Guangzhou and has received very high loads of nutrients from the drainage basin in the last three decades. As such, eutrophication has increasingly become an issue of concern (Huang et al., 2003; Ye et al., 2012). Dissolved inorganic nitrogen (DIN) concentrations in the PRE have increased approximately 4-fold from 1986 (19.3 µmol $L^{-1}$) to 2002 (76.1 µmol $L^{-1}$) (He and Yuan, 2007). This DIN increase has been attributed to increased inputs of domestic sewage, industrial wastewater, agricultural runoff and aquaculture in the watershed (Huang et al., 2003).

Recent observations based on monthly surveys between April 2010 and March 2011 and long term monitoring data from 1990 to 2014, have suggested that the lower PRE has emerged as a seasonal hypoxic zone (Qian et al., 2017). This is supported by our current study, as two relatively large hypoxic zones (> 300 $km^2$) were observed in the lower PRE with DO < 2 mg $L^{-1}$. However, the origin of the organic matter driving hypoxia in the lower PRE has not previously been examined. Here, we quantified the relative proportions of $OC_{mar}$ and $OC_{terr}$ contributing to DO drawdown in bottom waters of the lower PRE, an economically important coastal region. This study has important biological, societal and managerial implications for the region, particularly relating to water quality in the vicinity of Hong Kong in the lower PRE. For example, the government of Hong Kong is examining the efficacy of its costly Harbour Area Treatment Scheme project and if additional treatment should be implemented (http://www.gov.hk/en/residents/environment/water/harbourarea.htm).

**2 Materials and Methods**

**2.1 Sampling and analysis**

Interrupted by Typhoon Rammasun on 17-18 July 2014, our cruise was divided into two legs (Fig. 1). During Leg 1 on 13–16 July, we sampled Transects F4, F5 and Stations A08–A18. During Leg 2 on 19–27 July, we sampled Stations A01–A10, Transects F3 and F4, Stations A11–A17 and Transects F5, F6, F1, and F2, in sequence.

In order to monitor the development of hypoxia before and after the passage of the

typhoon, we revisited Station A10 three more times (13, 20 and 27 July).

According to the gauge in the upper Pearl River, water discharge peaked in June and

July. Typhoon Rammasun increased discharge during 15-18 July, with daily average

values of 19480, 26115, 22981 and 17540 $m^3$ $s^{-1}$, respectively. Nevertheless, the

freshwater discharge was 18908 $m^3$ $s^{-1}$ in leg 1 and 15698 $m^3$ $s^{-1}$ in leg 2, comparable

to the long-term (2000–2011) monthly average.

Temperature and salinity were determined with a SBE 25

Conductivity-Temperature-Depth/Pressure unit (Sea-Bird Co.). Water samples were

collected using 4 L Go-Flo bottles (General Oceanics). DIC and DO was measured at

all stations with depth profiles. Samples for $\delta^{13}C_{DIC}$ were collected primarily along

Transect A as well as at depth in low oxygen layers.

The DO concentrations in discrete water samples were measured on board within 8 h

using the classic Winkler titration method (Dai et al., 2006). In addition, we conducted

on-deck incubation experiments using unfiltered water taken from the hypoxic zone

on 27 July, 2014 following He et al. (2014). Bottom water from ~2 m above the

sediment surface was collected and incubated for 24 hours in 65 mL BOD bottles in

dark at ambient temperature controlled by the flowing surface water. Total oxygen

consumption rate was determined by comparing the DO concentration at the initial

and end point of the experiment.

DIC was measured with an infrared detector after acidifying 0.5–0.7 mL of water

sample with a precision of 0.1 % for estuarine and sea waters (Cai et al., 2004).

Dissolved calcium concentrations ($Ca^{2+}$) were determined using an EGTA titration

with a Metrohm 809 TITRANDO potentiometer, which has a precision better than $\pm 5$

$\mu mol\ kg^{-1}$ (Cao et al., 2011).

For $\delta^{13}C_{DIC}$ analysis, an ~20 mL DIC sample was converted into gaseous $CO_2$ and

progressively purified through a vacuum line. The pure $CO_2$ sample was analyzed with

an isotope ratio mass spectrometer (IRMS, Finnigan MAT 252, Bremen, Germany).

The analytical precision was better than 0.1 ‰.

Water samples for TSM (total suspended matter), POC and $\delta^{13}C_{POC}$ analysis were

concentrated onto preweighed and pre-combusted 0.7 $\mu m$ Whatman GF/F filters after

filtering 0.2–1.0 L of water under a mild vacuum (~ 25 kPa). Filters were washed with

distilled water and stored at -20 ℃. Prior to analysis, all filters were freeze-dried. TSM

was determined using the net weight increment on the filter and the filtration volume. Filters were decarbonated with 1.0 mol L$^{-1}$ HCl and dried at 40 ℃ for 48 h (Kao et al., 2012) and analyzed for POC and $\delta^{13}C_{POC}$ on an elemental analyzer coupled with an IRMS (EA-IRMS). The analytical precision for $\delta^{13}C_{POC}$ was better than 0.1 ‰. Chl-a was measured with a Turner fluorometer after extracting filters with 90 % acetone (He et al., 2010b). Calibrations were performed using a Sigma Chl-a standard.

**2.2 Three end-member mixing model**

We adopted a three end-member mixing model to construct the conservative mixing scheme among different water masses (Cao et al., 2011; Han et al., 2012):

$$F_{RI}+F_{SW}+F_{SUB}=1 \tag{1}$$

$$\theta_{RI}\times F_{RI}+\theta_{SW}\times F_{SW}+\theta_{SUB}\times F_{SUB}=\theta \tag{2}$$

$$S_{RI}\times F_{RI}+S_{SW}\times F_{SW}+S_{SUB}\times F_{SUB}=S \tag{3}$$

where $\theta$ and $S$ represent potential temperature and salinity; the subscripts RI, SW, and SUB denote the three different water masses (Pearl River plume water, offshore surface seawater and upwelled subsurface water); and $F_{RI}$, $F_{SW}$, and $F_{SUB}$ represent the fractions that each end-member contributes to the in situ samples. These fractions were applied to predict conservative concentrations of DIC (DIC$_{con}$) and its isotopic composition ($\delta^{13}C_{DICcon}$) resulting solely from conservative mixing.

$$DIC_{RI}\times F_{RI}+DIC_{SW}\times F_{SW}+DIC_{SUB}\times F_{SUB}=DIC_{con} \tag{4}$$

$$\frac{\delta^{13}C_{DICRI}\times DIC_{RI}\times F_{RI}+\delta^{13}C_{DICSW}\times DIC_{SW}\times F_{SW}+\delta^{13}C_{DICSUB}\times DIC_{SUB}\times F_{SUB}}{DIC_{con}}=\delta^{13}C_{DICcon} \tag{5}$$

The difference ($\Delta$) between measured and conservative DIC values represents the magnitude of the biological alteration of DIC (Wang et al., 2016).

**3 Results**

**3.1 Horizontal distribution**

Although the average freshwater discharge rate during our sampling period (16369 m$^3$ s$^{-1}$) was slightly higher than the multi-year (2000–2011) monthly average (15671 m$^3$ s$^{-1}$), typhoon Rammasun modified the system to some extent as shown from the evolution of chemical species at Station A10 before and after the typhoon (See Sect. 3.4). The interruption of Leg 1 due to the typhoon (July 17-18) led to a smaller survey area, covering only outside Lingdingyang Bay (traditionally regarded as the PRE),

while Leg 2 covered Lingdingyang Bay from the Humen Outlet to the adjacent coastal sea.

As depicted in Fig. 2, the sea surface temperature (SST) during Leg 1 (28.9-32.2 °C) was slightly higher than during Leg 2 (28.9-31.0 °C). Sea surface salinity (SSS) measurements showed that plume water was restricted more landward during Leg 2 than Leg 1. However, a steeper gradient to higher SST offshore during Leg 1 was likely induced by the upwelling of bottom water, featuring relatively high SSS (18.6), high DIC (1789 μmol kg$^{-1}$) and low DO saturation (DO%, 86 %). During Leg 1, the region with the most productivity was found east of the Wanshan Islands, characterized by high concentrations of Chl-a (8.0 μg kg$^{-1}$), low concentrations of DIC (1607 μmol kg$^{-1}$), and DO supersaturation, with the highest DO% greater than 160 % at Station F503. During Leg 2, there were three patches of high productivity, south of Huangmaohai, at the PRE entrance, and off Hong Kong. The central region of high productivity had the highest DO%, greater than 140% at Station A14, and was characterized by relatively high concentrations of Chl-a (7.8 μg kg$^{-1}$) and low concentrations of DIC (1737 μmol kg$^{-1}$).

As shown in Fig. 3, bottom water hypoxia during Leg 1 was located more centrally in the study area relative to the surface phytoplankton bloom. The center of the hypoxic zone was found at Station A10, characterized by the lowest observed DO concentrations (as low as 30 μmol kg$^{-1}$) and a relatively high concentration of DIC (2075 μmol kg$^{-1}$). During Leg 2, hypoxic conditions were no longer found at Station A10, and instead the largest hypoxic zone was discovered to the southwest of the Wanshan Islands, where the lowest DO values were observed (as low as 7 μmol kg$^{-1}$ at F304), and once again coincided with relatively high concentrations of DIC (2146 μmol kg$^{-1}$). We were unable to precisely constrain the areas of the regions impacted by bottom water hypoxia due to the limited spatial coverage, but our results suggest it covered an area of > 280 km$^2$ during Leg 1 and > 290 km$^2$ during Leg 2 according to the definition of hypoxia as DO < 2 mg L$^{-1}$ or 63 μM, or an area of > 900 km$^2$ during Leg 1 and > 800 km$^2$ during Leg 2 assuming the threshold of the oxygen-deficit zone was < 3 mg L$^{-1}$ or 95 μM (Rabalais et al., 2010; Zhao et al., 2017).

## 3.2 Vertical distribution

During Leg 1, plume water reached 50 km offshore from the entrance of the PRE, forming a 5–10 m thick surface layer (Fig. 4b). Both the thermocline and halocline contributed to the stability of the water column structure, which favored the formation of bottom water hypoxia. The thickness of the bottom water hypoxic layer was ~ 5 m. The region of highest productivity, however, was not observed in the same location as the hypoxic zone, but further offshore.

During Leg 2, although the passing of the typhoon would be expected to absorb large amounts of potential heat and cause extensive mixing of the water column, the enhanced freshwater discharge could rapidly re-stratify the water column and facilitate the re-formation of hypoxia. This time, the primary region of hypoxia was observed directly below the bloom, with a thickness of 3 m (Fig. 4i). Additionally, near the Humen Outlet we observed low DIC (1466 $\mu$mol kg$^{-1}$) and moderately low DO (89 $\mu$mol kg$^{-1}$), which reflected the input of the low DO water mass from upstream as reported previously (Dai et al., 2006; Dai et al., 2008a; He et al., 2014).

## 3.3 Isotopic composition of DIC and POC

The $\delta^{13}C$ values of DIC became progressively heavier from stations dominated by freshwater (~ -11.4 ‰) to off-shore seawater (~ -0.6 ‰), with a relatively wide range of values beyond a salinity of 13 (Fig. 5). Owing to a malfunction of the instrument, $\delta^{13}C_{POC}$ data from our cruise were not available. Instead, we reported a valid $\delta^{13}C_{POC}$ dataset from a 2015 summer cruise in approximately the same region. $\delta^{13}C_{POC}$ values showed a similar trend with $\delta^{13}C_{DIC}$, i.e. $^{13}C$ enriched seaward, from ~ -28 ‰ to ~ -20 ‰. In the bloom, where the DO% was above 125 %, the mean $\delta^{13}C$ value for POC was -19.4±0.8 ‰ (n=8), which was within the typical range of marine phytoplankton (Peterson and Fry, 1987). As shown in Fig. 5, there was a large $\delta^{13}C_{POC}$ decrease near a salinity of 15. Geographically, it was located at the mixing dominated zone in inner Lingdingyang Bay, where intense resuspension of $^{13}C$ depleted sediments may occur (Guo et al., 2009).

## 3.4 Reinstatement of the hypoxic station after Typhoon Rammasun

Typhoon Rammasun made landfall at Zhanjiang, located 400 km to the southwest of the PRE, at 20:00 LT (Local Time) on 18 July, and was dissipated by 05:00 LT on 20

July. The typhoon completely de-stratified the water column during its passing. However, the associated heavy precipitation and runoff appeared to re-establish stratification rather quickly, within one day, as suggested by the salinity gradient (18–30) from 0–10 m depth during Leg 2 at 15:20 LT on 20 July (Fig. 6b). In order to capture the evolution of DO between the disruption and reinstatement of stratification, we resumed our cruise and revisited Station A10 (Fig. 6). On 13 July, the bottom water at Station A10 was the hypoxic core, with the lowest observed DO (30 µmol kg$^{-1}$) and highest DIC (2075 µmol kg$^{-1}$) concentrations. On 20 July, the results showed that the temperature homogeneous layer in the bottom water (9–13 m) might reflect the remnants of typhoon-induced mixing (Fig. 6a), while the reduction in salinity at <9 m depicted the rapid re-establishment of stratification as a result of enhanced freshwater discharge (Fig. 6b). Bottom water DO increased to 153 µmol kg$^{-1}$ and DIC decreased to 1901 µmol kg$^{-1}$ as a result of the typhoon-induced water column mixing and aeration. In addition, TSM increased sharply from 20.2 before the typhoon to 36.6 mg kg$^{-1}$, suggesting large volumes of sediment had been resuspensed during its passing. On 27 July, one week after the typhoon, strong thermohaline stratification was re-established in the whole water column. Along with the intensifying stratification, bottom water DO decreased to 99 µmol kg$^{-1}$ indicating continuous DO depletion and the potential for hypoxia formation. Meanwhile, bottom water DIC concentrations increased to 2000 µmol kg$^{-1}$ and dissolved inorganic phosphate (DIP) rose from 0.28 to 0.57 µmol kg$^{-1}$. Moreover, bottom-water TSM returned to pre-typhoon (13 July) levels.

**4 Discussion**

**4.1 Selection of end-members and model validation**

The potential temperature-salinity plot displayed a three end-member mixing scheme over the PRE and adjacent coastal waters (Fig. 7a), consisting of Pearl River plume water, offshore surface seawater and upwelled subsurface water. During the summer, a DIC concentration of ~1917 µmol kg$^{-1}$ was observed at S=33.7, which can be regarded as the offshore surface seawater end-member (Guo and Wong, 2015). Here, by choosing S=34.6 as the offshore subsurface water salinity end-member, we obtained a DIC value of ~2023 µmol kg$^{-1}$, similar to the value at ~100 m depth adopted by Guo and Wong (2015). For the plume end-member, it was difficult to directly select from the

field data, because biological alteration might lead to altered values within the plume influenced area. Therefore, we first assumed that the plume water observed on the shelf consisted of a mixture of freshwater and offshore surface seawater. Then, we compiled 3 years of surface data from the summer (August 2012, July 2014 and July 2015) to extrapolate the relatively stable freshwater end-member and examine the biological effect on DIC-salinity relationships. By constraining DIC end-members (freshwater and offshore surface seawater), we observed that DIC remained overall conservative when salinity was <10.8 but showed removal when salinity was >10.8 (Han et al., 2012). Thus, we derived plume end-member values (1670±50 μmol kg$^{-1}$) from the DIC-salinity conservative mixing curve at S=10.8. Furthermore, S=10.8 was observed at the innermost station (A08) during Leg 1, which agreed well with the spatial and temporal scale of the actual water mass mixing in our survey. To confirm our results, we also used a freshwater end-member (S=0), but the output of the model showed little difference from that based on the plume end-member at S=10.8.

The $\delta^{13}C_{DIC}$ value was 0.6±0.2 ‰ in the offshore surface seawater at S=~33.7, where nutrient ($NO_3^-$+$NO_2^-$ and DIP) concentrations were close to their detection limits and DO was nearly saturated, indicating little biological activity. As DIC remained overall conservative when salinity was < 10.8, the $\delta^{13}C_{DIC}$ value of -11.4±0.2 ‰ at S < 0.4 is representative of the freshwater source. Assuming the plume water is a mixture of freshwater and offshore surface seawater, the initial plume end-member of $\delta^{13}C_{DIC}$ at S=10.8 can be calculated via an isotopic mass balance (-7.0±0.8 ‰). A summary of the end-member values used in this study is listed in Table 1.

We calculated the fractions of the three water masses based on potential temperature and salinity equations, so as to predict conservative DIC ($DIC_{con}$) and its isotopic composition ($\delta^{13}C_{DICcon}$) solely from conservative mixing. We chose the concentration of $Ca^{2+}$ as a conservative tracer to validate our model prediction, assuming $CaCO_3$ precipitation or dissolution is not significant. This assumption is supported by a strong linear relationship between surface water $Ca^{2+}$ and salinity, and aragonite oversaturation ($\Omega_{arag}$=2.6±0.7) in the subsurface water. Our model derived values were in good accordance with the field-observed values (Fig. 7b), which strongly supported our model prediction.

As shown in Fig. 7c, most of the observed DIC concentrations in the subsurface water were higher than the conservative values, as a result of DIC production via OC

oxidation. This coincided with lighter $\delta^{13}C_{DIC}$ values than conservative, owing to the accumulation of isotopically lighter carbon entering the DIC pool from remineralized organic matter (Fig. 7d). Based on the differences between the observed and conservative values of DIC and $\delta^{13}C_{DIC}$, the carbon isotopic composition of the oxygen-consuming organic matter could be traced precisely (see details in Sect. 4.2).

In the subsurface water, the bulk of $\Delta$DIC values varied from 0 to 132 $\mu$mol kg$^{-1}$, coupled with a range of apparent oxygen utilization (AOU) values from 0 to 179 $\mu$mol kg$^{-1}$. $\Delta$DIC values positively correlated with AOU (Fig. 7e), corresponding to the fact that the additional DIC was supplied by organic matter remineralization via aerobic respiration. The slope of $\Delta$DIC vs. AOU in the subsurface water was 0.71$\pm$0.03, which agrees well with classic Redfield stoichiometry (i.e., 106/138=0.77), providing further evidence for aerobic respiration as the source of added DIC. As a first order comparison, the water column total oxygen consumption rate of 9.8 $\mu$mol L$^{-1}$ d$^{-1}$ could well support the oxygen decline rate observed at Station A10 in the hypoxic zone between 20 July and 27 July (Fig. 6), which was 7.7 $\mu$mol L$^{-1}$ d$^{-1}$. This comparison along with the stoichiometry between $\Delta$DIC and AOU strongly suggests that water column aerobic respiration may be predominate in the formation of the hypoxia in the present case.

**4.2 Isotopic composition of the oxygen-consuming OC**

The DIC isotopic mass balance is shown in Eq. (6) (Wang et al., 2016):

$$\delta^{13}C_{DICobs} \times DIC_{obs} = \delta^{13}C_{DICcon} \times DIC_{con} + \delta^{13}C_{DICbio} \times DIC_{bio} \tag{6}$$

where the subscripts obs, con and bio refer to the field-observed, conservative and biologically altered values.

Degradation of OC typically produces DIC with minor isotopic fractionation from the OC substrate (Hullar et al., 1996; Breteler et al., 2002). Thus, the isotopic composition of DIC$_{bio}$ (i.e., $\delta^{13}C_{DICbio}$) should be identical to the $\delta^{13}C$ of the OC ($\delta^{13}C_{OCx}$), which consumed oxygen and produced DIC$_{bio}$. $\delta^{13}C_{OCx}$ was derived from the mass balance equations of both DIC and its stable isotope:

$$\delta^{13}C_{OCx} = \frac{\delta^{13}C_{obs} \times DIC_{obs} - \delta^{13}C_{con} \times DIC_{con}}{DIC_{obs} - DIC_{con}} \tag{7}$$

Equation (7) can be rearranged into Eq. (8):

$$\Delta(\delta^{13}C_{DIC} \times DIC) = \delta^{13}C_{OCx} \times \Delta DIC \tag{8}$$

As shown in Fig. 8, the slope of the linear regression represents $\delta^{13}C_{OCx}$ or $\delta^{13}C_{DICbio}$, which here is equal to -23.2±1.1 ‰. This value reflects the original $\delta^{13}C$ signature of the remineralized organic matter contributing to the observed addition of DIC.

Although studies have shown selective diagenesis of isotopically heavy or light pools of organic matter (Marthur et al., 1992; Lehmann et al., 2002), these effects are small compared to the isotopic differences among different sources of organic matter (Meyers, 1997). It is thus reasonable to assume that the isotopic ratios are conservative and that physical mixing of the end-member sources determine the isotopic composition of organic matter in natural systems (Gearing et al., 1984; Cifuentes et al., 1988; Thornton and McManus, 1994). The relative contributions of marine and terrestrial sources to oxygen-consuming organic matter in our study area could be estimated based on the following equation (Shultz and Calder, 1976; Hu et al., 2006):

$$f(\%)=\frac{\delta^{13}C_{mar}-\delta^{13}C_{OCx}}{\delta^{13}C_{mar}-\delta^{13}C_{terr}}\times100 \tag{9}$$

Here, for the terrestrial end-member ($\delta^{13}C_{terr}$), we adopted the average $\delta^{13}C$ value of POC sampled near the Humen Outlet (S<4), which represents the predominant source of riverine material entering the estuary (He et al., 2010b). The mean $\delta^{13}C_{POC}$ value, -28.3±0.7 ‰ (n=7), is very similar to the freshwater $\delta^{13}C_{POC}$ value of -28.7 ‰ reported by Yu et al. (2010), which reflected a terrigenous mixture of C3 plant fragments and forest soils. For the marine end-member ($\delta^{13}C_{mar}$), we calculated the mean surface water $\delta^{13}C_{POC}$ value (-19.4±0.8 ‰, n=8) from stations with S>26 where significant phytoplankton blooms were observed, as indicated by DO supersaturation (DO% > 125 %) and relatively high pH values (> 8.3) and POC contents (5.3±2.4 %). This value is similar, although slightly heavier than the marine end-member used by Chen et al. (2008), who measured a $\delta^{13}C$ value of -20.9 ‰ in tow-net phytoplankton samples from outer Lingdingyang Bay, in the same region as this study. Additionally, He et al. (2010a) reported a $\delta^{13}C$ value of -20.8±0.4 ‰ in phytoplankton collected from the northern South China Sea. These values are consistent enough for us to compile and use an average $\delta^{13}C_{mar}$ value of -20.5±0.9 ‰. This value agrees well with the reported stable carbon isotopic signature of marine organic matter in other coastal regions. For example, mean isotopic values of phytoplankton were reported as -20.3±0.6 ‰ in Narragansett Bay (Gearing et al., 1984), -20.3±0.9 ‰ in Auke Bay

and Fritz Cove (Goering et al., 1990), and -20.1±0.8 ‰ in the Gulf of Lions

(Harmelin-Vivien et al., 2008).

Our model results suggest that marine organic matter contributed to 65±16 % of the

observed oxygen consumption, while terrestrial organic matter accounted for the

remaining 35±16 %. It is thus clear that marine organic matter from

eutrophication-induced primary production dominated oxygen consumption in the

hypoxic zone; however, terrestrial organic matter also contributed significantly to the

formation and maintenance of hypoxia in the lower PRE and adjacent coastal waters.

**4.3 Comparison with hypoxia in the East China Sea off the Changjiang Estuary**

As one of the largest rivers in the world, the Changjiang has been suffering from

eutrophication for the past few decades (Zhang et al., 1999; Wang et al., 2014). In

summer, sharp density gradients with frequent algal blooms and subsequent organic

matter decomposition cause seasonal hypoxia in the bottom water of the ECS off the

CJE. Wang et al. (2016) revealed that the remineralization of marine organic matter

($OC_{mar}$) overwhelmingly (nearly 100 %) contributed to DO consumption in the ECS off

the CJE. However, our present study showed that less $OC_{mar}$ contributed to the oxygen

depletion (65±16 %) in the hypoxic zone of the lower PRE.

As shown in Fig. 5, there is little difference between $\delta^{13}C_{DIC}$ and $\delta^{13}C_{POC}$ values of

the marine end-member. However, the $\delta^{13}C_{DIC}$ and $\delta^{13}C_{POC}$ values of the freshwater

end-member showed some dissimilarity, with lighter values in the PRE (-11.4±0.2 ‰,

21 -28.3±0.7 ‰) than in the CJE (-8.8 ‰, -24.4±0.2 ‰). In Fig. 7e, the amplitude of

22 ΔDIC and AOU values suggest a similar intensity of OM biodegradation, and the

23 slope of ΔDIC vs. AOU (0.71±0.03 vs. 0.65±0.04) indicates a predominance of

24 aerobic respiration in the two systems. As seen from Table 2, there is no significant

difference between the $\delta^{13}C$ values of surface sediments within the hypoxic zones of

the PRE and CJE. However, data in Fig. 7a show generally higher water temperatures

in the PRE than in the CJE. For instance, the temperature of surface and subsurface

seawater end-members in the PRE is 2-3 ℃ higher than in the CJE. From a spatial

point of view, the distance from the river mouth to the hypoxic zone in the CJE is 2-3

30 times longer than in the PRE, possibly resulting in a longer travel time of $OC_{terr}$.

Therefore, we contend that the difference in the predicted distributions of marine and

terrestrial sources of organic matter contributing to oxygen consumption in and off the

PRE and CJE is likely related to differences in the bioavailability of $OC_{terr}$ and $OC_{mar}$, the microbial community structures and the physical settings between these two hypoxic systems.

Although C3 plants dominate and C4 plants are minor in both the Pearl River and Changjiang drainage basins (Hu et al., 2006; Zhu et al., 2011a), the $OC_{terr}$ delivered from these two watersheds experiences varying degrees of degradation within the estuaries before being transported into the coastal hypoxic zones. In the CJE, approximately 50 % of $OC_{terr}$ becomes remineralized during transport through the estuary, likely due to efficient OM unloading from mineral surfaces (Zhu et al., 2011a) and longer residence times within the estuary, facilitating microbial transformation and degradation. In contrast, the PRE appears to be a somewhat intermediate site with the export of $OC_{terr}$ being closely associated with sedimentary regimes and not characterized by extensive degradative loss (Strong et al., 2012). Thus, the bioavailability of $OC_{terr}$ that reached the hypoxic zone is likely higher in the PRE than in the CJE. Moreover, the increased precipitation and runoff during the typhoon may have mobilized additional fresh anthropogenic OM from surrounding megacities (e.g. Guangzhou, Shenzhen and Zhuhai) deposited in the river channel, which could lead to more labile $OC_{terr}$ in the PRE. Additionally, the difference in bacterial community structure between the two systems may have played a role. Recent studies have demonstrated that the bacterial community in the PRE is characterized by higher relative abundances of Actinobacteria and lower relative abundances of Cytophaga-Flavobacteria-Bacteroides (CFB) than in the CJE (Liu et al., 2012; Zhang et al., 2016). Whether such differences would promote the degradation of $OC_{terr}$ in the PRE relative to the CJE remains unknown. Finally, the temperature of the bottom water in the PRE hypoxic zone (27–29 ℃) was higher than in the CJE hypoxic zone (21.5–24.0 ℃), which may have accelerated the rates of bacterial growth and OM decomposition (Brown et al., 2004).

**5 Conclusions**

Based on a three end-member mixing model and the mass balance of DIC and its isotopic composition, we demonstrated that the organic matter decomposed via aerobic respiration in the stratified subsurface waters of the lower PRE and adjacent coastal waters was predominantly $OC_{mar}$ (49-81 %, mean 65 %), with a significant portion of

OC$_{terr}$ also decomposed (19-51 %, mean 35 %). The relative distribution of organic matter sources contributing to oxygen drawdown differs in the hypoxic zone off the CJE, where it is caused almost entirely by OC$_{mar}$. These differences have important implications for better understanding the controls on hypoxia and its mitigation. Nevertheless, with respect to increasing coastal nutrient levels, a significant implication of the present study is that reducing and managing nutrients is critical to control eutrophication and, subsequently, to mitigate hypoxia (Conley et al., 2009; Paerl, 2009; Mercedes et al., 2015; Stefan et al., 2016). Given that OC$_{terr}$ also contributes to the consumption of oxygen in the lower PRE hypoxic zone, it is crucial to characterize the source of this oxygen-consuming terrestrial organic matter, whether from natural soil leaching and/or anthropogenic wastewater discharge, so as to make proper policies for hypoxia remediation.

The processes involved in the partitioning of organic matter sources, their isotopic signals and their subsequent biogeochemical transformations in the PRE hypoxic zone are illustrated in the conceptual diagram in Fig. 9. The river delivers a significant amount of nutrients and terrestrial organic matter to the estuary, stimulating phytoplankton blooms in the surface water at the lower reaches of the estuary where turbidity is relatively low and conditions are favourable for phytoplankton growth (Gaston et al., 2006; Dai et al., 2008b; Guo et al., 2009). The subsequent sinking of this biomass along with terrestrial organic matter below the pycnocline consumes oxygen and adds respired DIC to subsurface waters, resulting in coastal hypoxia. Therefore, we conclude that within the PRE and adjacent coastal areas, the most important biological process with respect to forming and maintaining hypoxic conditions is aerobic respiration.

*Acknowledgments*. This research was funded by the National Natural Science Foundation of China through grants 41130857, 41576085 and 41361164001. We thank Tengxiang Xie, Li Ma, Shengyao Sun, Chenhe Zheng and Liangrong Zou for their assistance in sample collections; Yan Li and Yawen Wei for providing the calcium concentration data; Liguo Guo, Tao Huang and Dawei Li for assisting on the measurements of DIC, nutrients and $\delta^{13}$C$_{POC}$. The captain and the crew of R/V *Kediao 8*

are acknowledged for their cooperation during the cruise. Finally, we express our gratitude to three anonymous referees for their insightful and constructive comments and input.

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

**Table 1.** Summary of end-member values and their uncertainties adopted in the three end-member mixing model.

| Water Mass | $\theta(°C)$ | Salinity | DIC ($\mu mol\ kg^{-1}$) | $\delta^{13}C_{DIC}$ (‰) | $Ca^{2+}$ ($\mu mol\ kg^{-1}$) |
|---|---|---|---|---|---|
| Plume | 30.6±1.0 | 10.8 | 1670±50[a] | -7.0±0.8[b] | 3670±16[c] |
| Surface | 31.0±1.0 | 33.7±0.2 | 1917±3 | 0.6±0.2 | 9776±132[c] |
| Subsurface | 21.8±1.0 | 34.6±0.1 | 2023±6 | 0.1±0.1 | 10053 |

[a] In order to derive a proper plume end-member value, we took advantage of 3 years of surface dataset from summer cruises (see Sect. 4.1). For DIC, the data is from cruises during August 2012, July 2014 and July 2015.

[b] See details in Sect. 4.1.

[c] The $Ca^{2+}$ values of the plume and surface seawater end-member are derived from a conservative mixing calculation ($Ca^{2+}$ vs. S) based on 3 years of surface data during the summer (August 2012, July 2014 and July 2015).

1 **Table 2.** Comparison of $\delta^{13}$C values in surface sediments within the hypoxic zone[a] between

2 the PRE and CJE.

| $\delta^{13}$C (‰) | Mean$\pm$SD | Stations involved | References |
|---|---|---|---|
| | | Pearl River Estuary | |
| -23.4 ~ -22.1 | -22.9$\pm$0.5 | A4, A5, C1-C4, D1 | Hu et al. 2006 |
| -23.2 ~ -22.3 | -22.7$\pm$0.5 | 28, 29, 30 | Zong et al. 2006 |
| -23.6 ~ -21.5 | -22.5$\pm$1.1 | E8-1, E7A, S7-1, S7-2 | He et al. 2010a |
| -[b] | -23.1$\pm$0.6 | Clustering groups G6 and G7 | Yu et al. 2010 |
| Average | -22.8$\pm$0.6 | | |
| | | Changjiang Estuary | |
| -22.9 ~ -20.9 | -21.8$\pm$0.6 | -[c] | Tan et al. 1991 |
| -22.4 ~ -19.9 | -21.2$\pm$1.0 | 32, 37, 38, 42, 48, 49, 54, 56, 64 | Kao et al. 2003 |
| -22.7 ~ -20.8 | -22.0$\pm$0.8 | H1-12, H2-10, H2-11, S1-2, S2-4 | Xing et al. 2011 |
| -23.5 ~ -20.4 | -22.6$\pm$1.0 | 3, 12, 13, 20-25 | Yao et al. 2014 |
| Average | -21.9$\pm$1.0 | | |

[a]In the PRE, the data is from similar sites to our present study, which is in the northeast (Leg 1)

and southwest (Leg 2) of the Wanshan Islands. While in the CJE, the hypoxic zone is located

around 30.0 °N–32.0 °N, 122.7 °E–123.2 °E, which is frequently reported in previous studies

(Li et al., 2002; Zhu et al., 2011b; Wang et al., 2016).

[b]The authors provide an average value of clustering groups instead of individual data from each

site.

[c]In Fig. 7 of Tan et al. (1991), the sampling sites are shown without numbers.

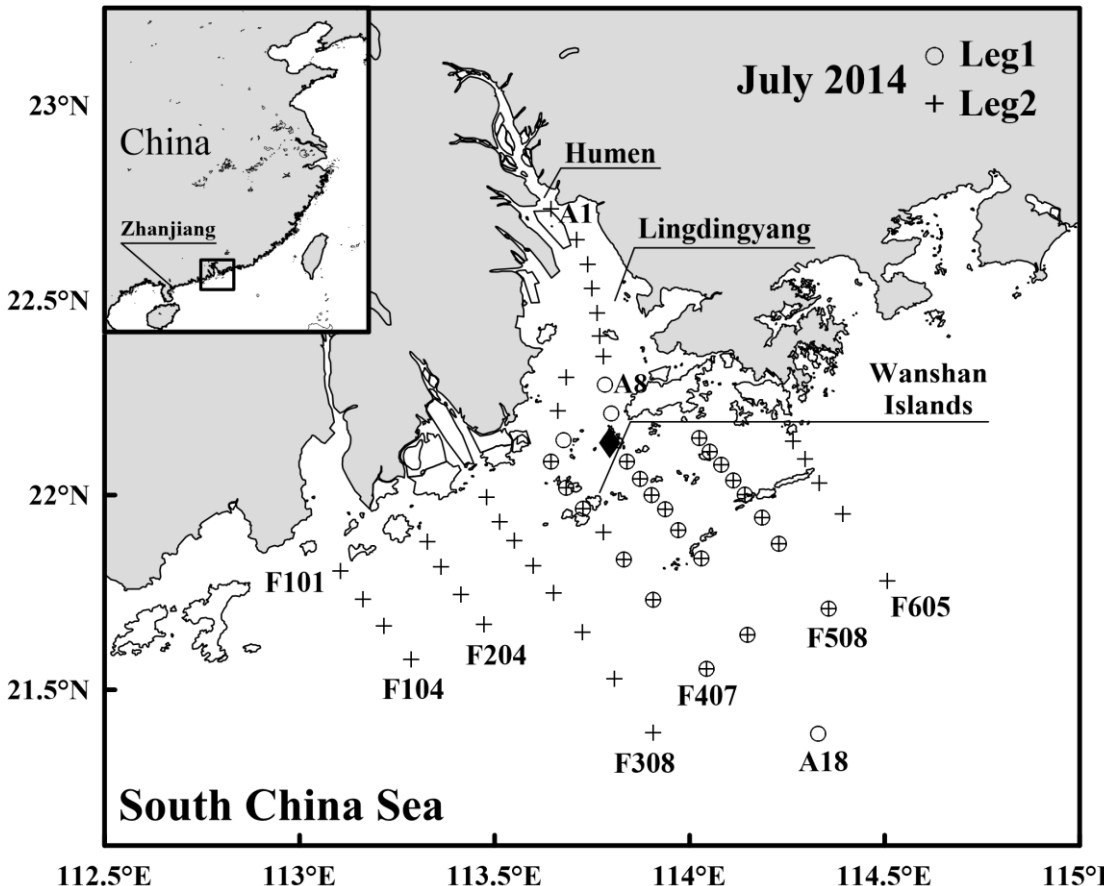

Figure 1. Map of the Pearl River Estuary and adjacent coastal waters. The open circles denote
Leg 1 stations visited on 13–16 July 2014, and the crosses represent Leg 2 stations visited on
19–27 July 2014. Note that the filled diamond is the location of Station A10.

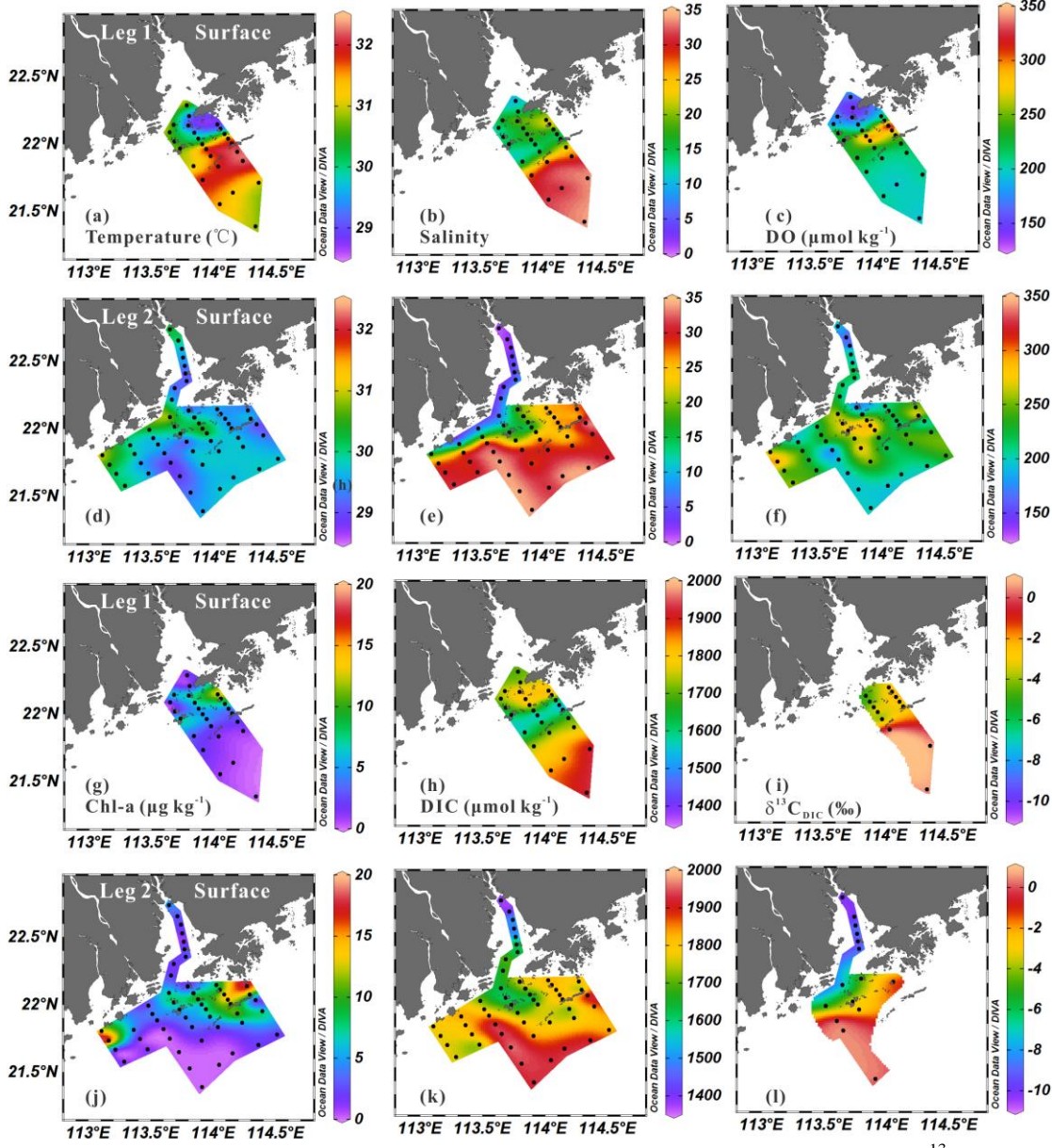

Figure 2. Surface water distribution of temperature, salinity, DO, Chl-a, DIC and $\delta^{13}C_{DIC}$ during Leg 1 (a–c, g–i) and Leg 2 (d–f, j–l).

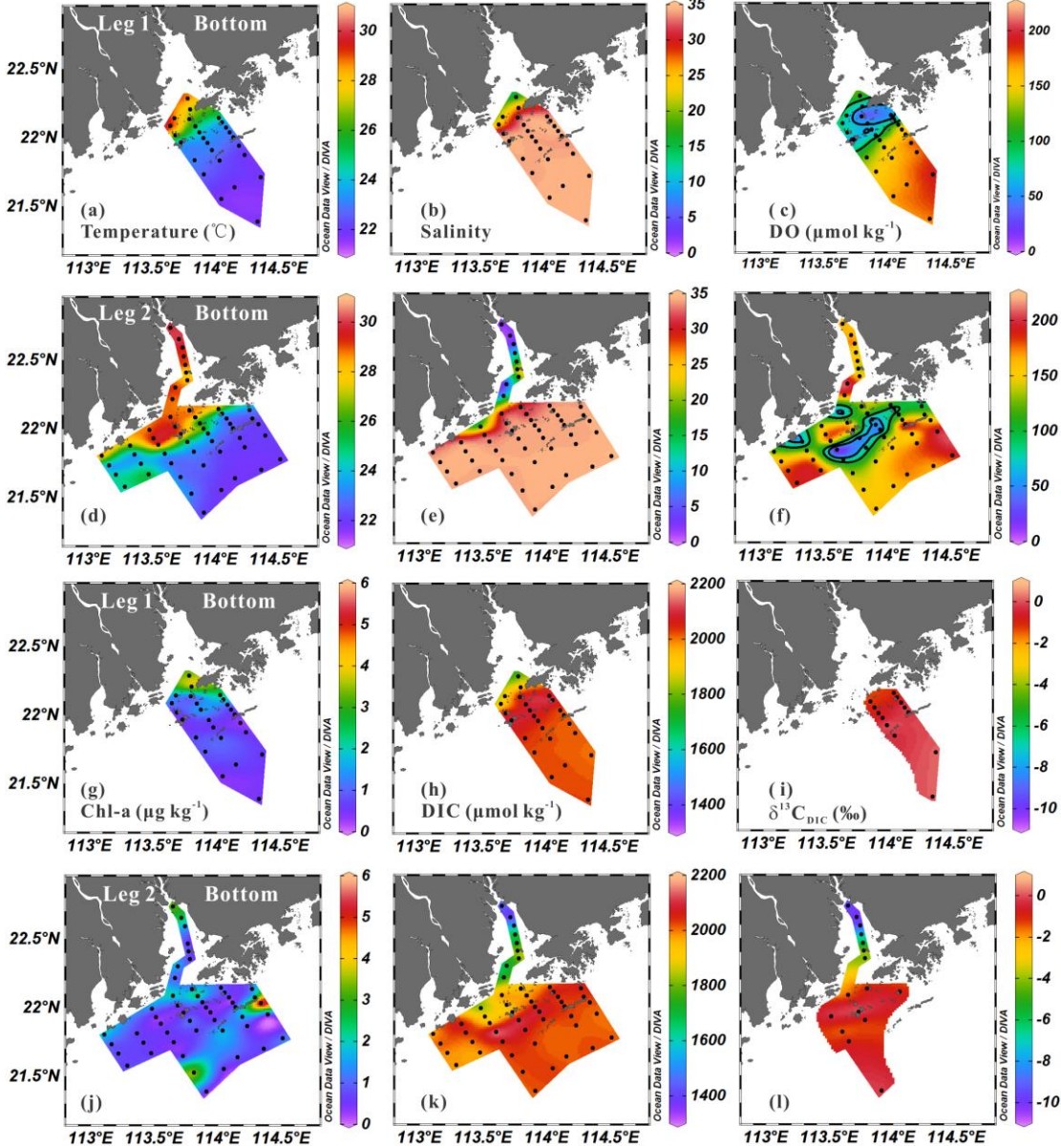

**Figure 3.** Bottom water distribution of temperature, salinity, DO, Chl-a, DIC and $\delta^{13}C_{DIC}$

during Leg 1 (a–c, g–i) and Leg 2 (d–f, j–l). Note that the black lines in (c) and (f) indicate DO

contours of 63 μM and 95 μM.

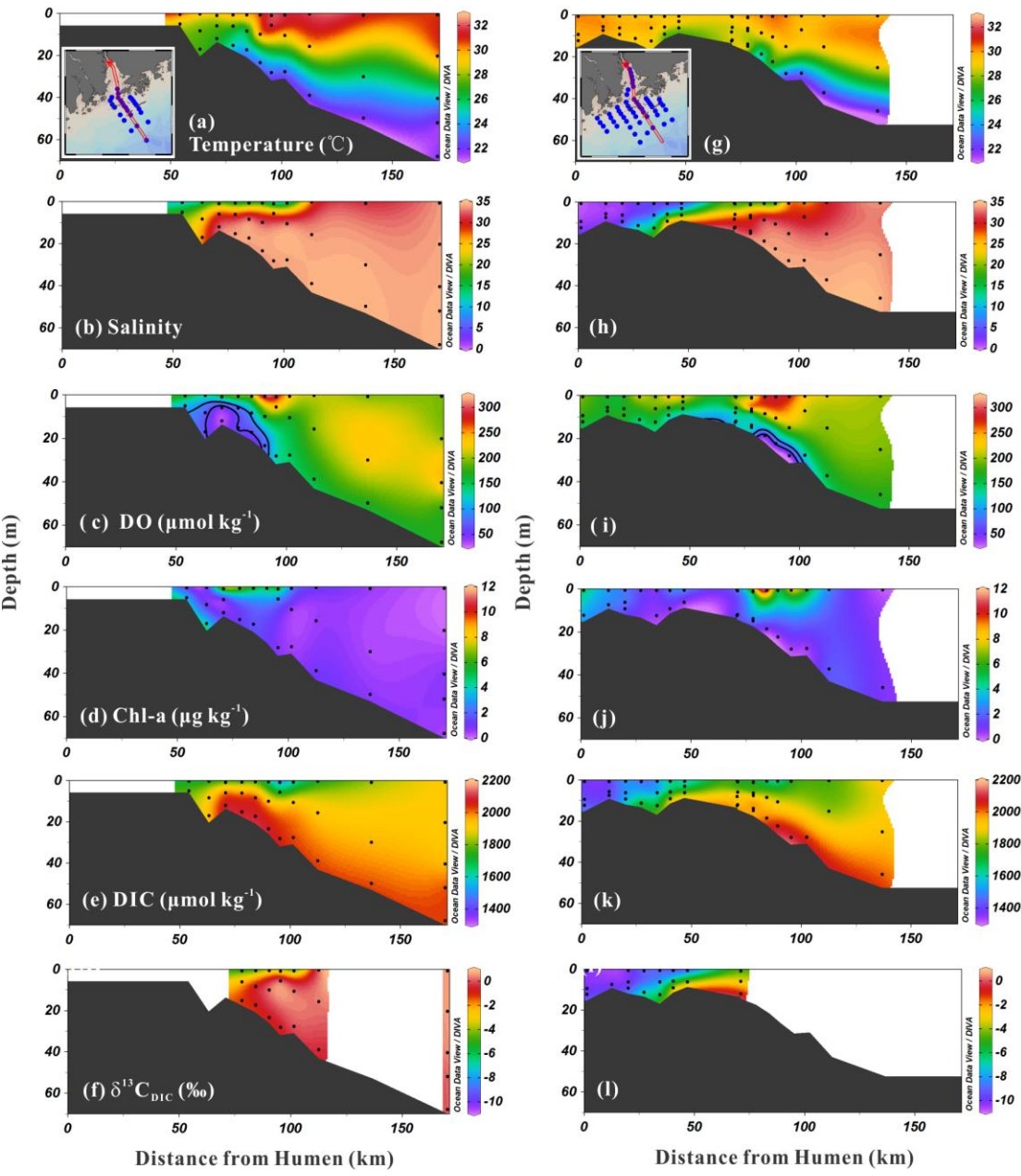

**Figure 4.** Profiles of temperature, salinity, DO, Chl-a, DIC and δ¹³C$_{DIC}$ along Transect A during Leg 1 (a–f) and Leg 2 (g–l). Note that the black lines in (c) and (i) indicate DO contours of 63 μM and 95 μM.

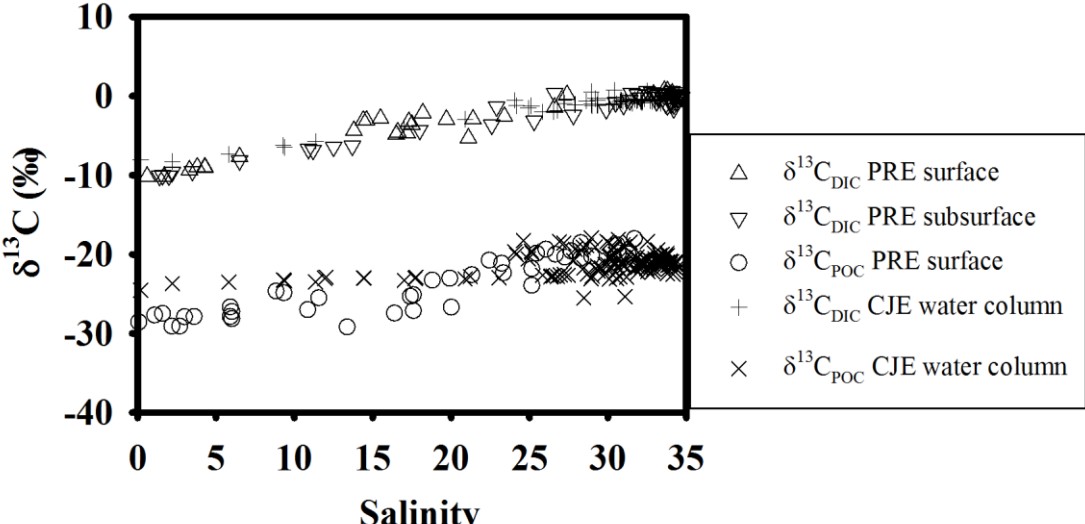

**Figure 5.** Distribution of $\delta^{13}C_{DIC}$ and $\delta^{13}C_{POC}$ with respect to salinity in the PRE. The up-facing and down-facing triangles denote surface and subsurface $\delta^{13}C_{DIC}$ data, respectively, from July 2014, while the open circles represent $\delta^{13}C_{POC}$ values in surface water from July 2015. Additionally, the plus signs and crosses show the $\delta^{13}C_{DIC}$ and $\delta^{13}C_{POC}$ data, respectively, from the Changjiang Estuary (CJE) in Wang et al. (2016).

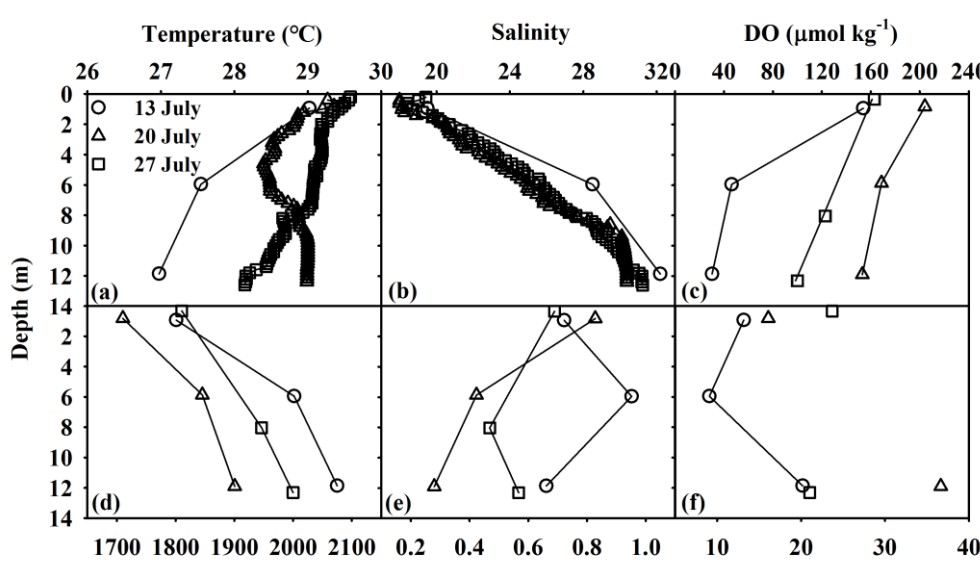

Figure 6. Profiles of (a) temperature, (b) salinity, (c) DO, (d) DIC, (e) DIP, (f) TSM and their

evolution during repeated sampling at Station A10.

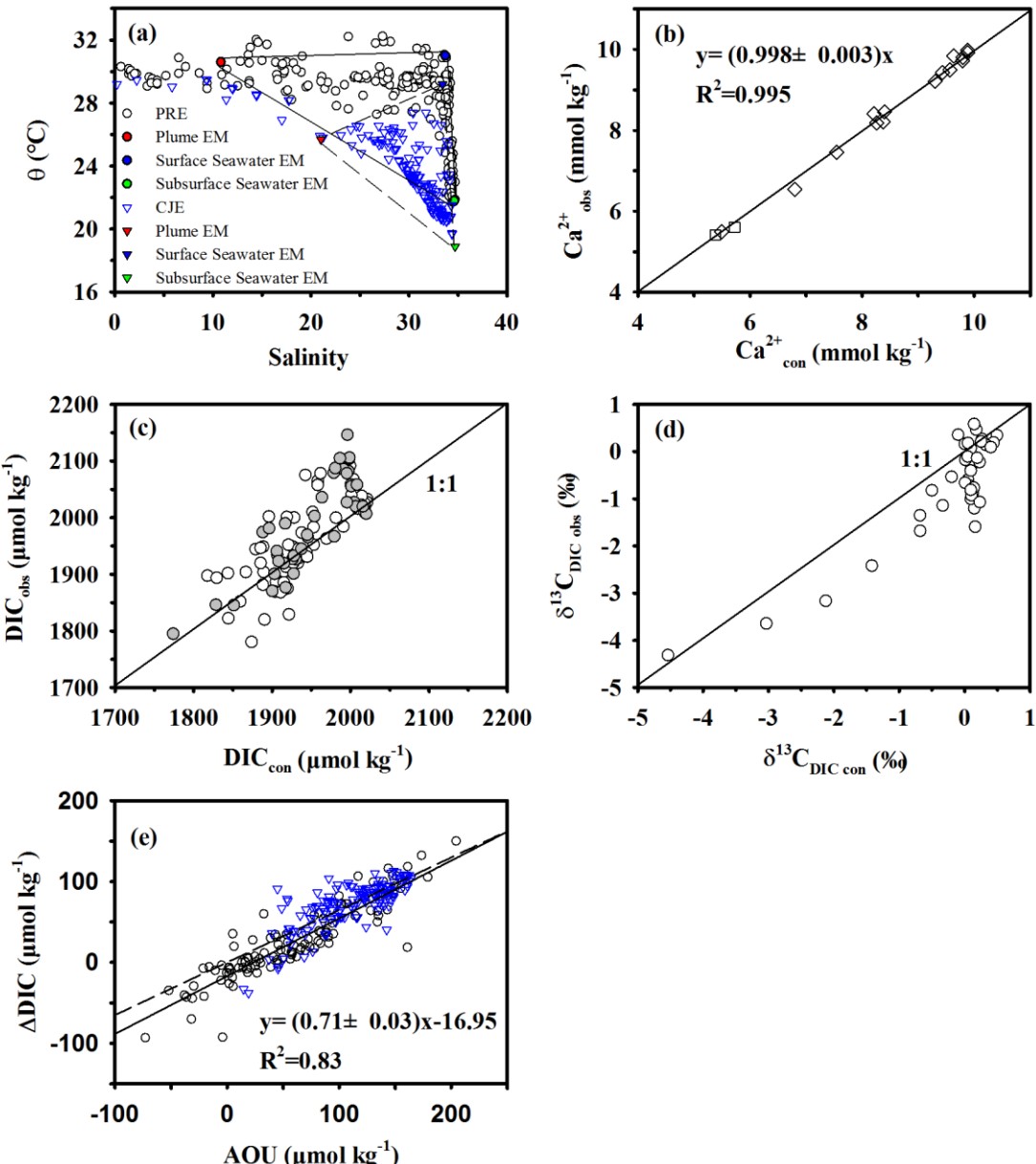

**Figure 7.** (a) Potential temperature (θ) ( ℃) vs. salinity in the PRE and adjacent coastal waters

(open circles) based on data collected during the July 2014 cruise. The three end-members are

shown as different coloured symbols. The blue triangles represent data collected during the

August 2011 cruise in the Changjiang Estuary (CJE) (Wang et al., 2016); (b) Correlation

between the field-observed $Ca^{2+}$ ($Ca^{2+}_{obs}$) and conservative $Ca^{2+}$ ($Ca^{2+}_{con}$). The straight line

denotes a linear regression line of both surface (square) and subsurface (diamond) data; (c), (d)

Relationship between observed and conservative DIC and $\delta^{13}C_{DIC}$ values. The straight line

represents a 1:1 reference line. Note that the grey dots in Fig. 7c identify data also in Fig. 7d;

and (e) Correlation of ΔDIC vs. AOU for all subsurface water data. ΔDIC is the difference

between the field-observed and conservative DIC concentrations. Also shown is the data from

1  Wang et al. (2016). The straight and dashed lines indicate linear regressions of data from the

2  PRE and CJE, respectively.

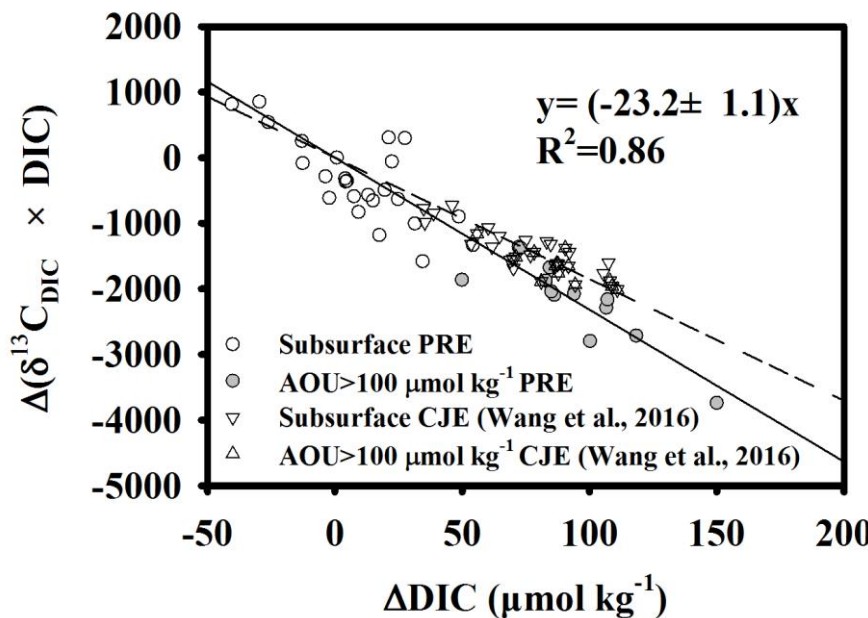

**Figure 8.** $\Delta$ ($\delta^{13}C_{DIC} \times DIC$) vs. $\Delta DIC$ in the PRE. Samples were collected from subsurface

water (> 5 m). The grey circles represent samples with AOU > 100 μmol kg$^{-1}$. $\Delta$ is the

difference between the field-observed and conservative values. Also shown is data from the

Changjiang Estuary (CJE) reported by Wang et al. (2016). The straight and dashed lines

indicate linear regression lines of data from the PRE and CJE, respectively.

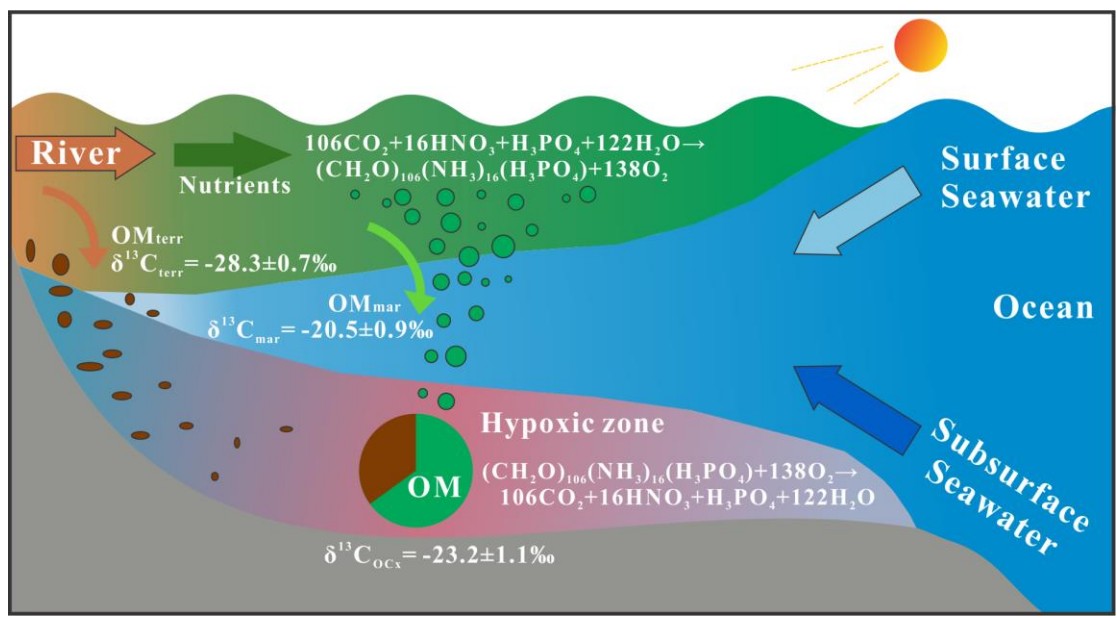

2  **Figure 9.** A conceptual diagram illustrating the partitioning of oxygen-consuming organic

3  matter ($OC_{mar}$ vs. $OC_{terr}$) within the hypoxic zone in the lower PRE and the adjacent coastal

4  area. See Sect. 5 for explanations.

