# Peer review of "Tracing the origin of the oxygen-consuming 1"

_Biogeosciences, 2017_

## Referee Comment (RC1) · Anonymous Referee #1 · 22 Mar 2017

The paper by Su et al. deals with hypoxia and its causes in the Pearl River Estuary (PRE). Hypoxic events are a growing concern from human societies as they threat the environment and the resources on which coastal population lives. Furthermore, it strongly impacts the environment with a resilience time which is now largely unknown. Attribution of hypoxia to a type of organic matter mineralization has been seldom done, and is very interesting from a watershed manager point of view. It is the kind of effort that Su and his/her colleague have started, and for that reason their paper is of potential great interest. Indeed, they use DIC stable isotopic composition and a wealth of other data collected from two cruises in the Pearl River estuary and the adjacent coastal

zone to estimate the contribution of terrestrial and marine organic matter to the decrease of oxygen in stratified bottom waters. The contribution is thus original as only a few attempts exist to quantify the source, e.g. in the South China Sea near Changjiang River, but the reading of the paper raises some fundamental issues which are poorly answered in the manuscript: was the perturbation of the typhoon small enough that it can be neglected (see page 4, line 17-18)? Is the proportion of 75-25% a robust estimate of the contribution of the marine and terrestrial pools? Can we quantify better the uncertainty? Overall the paper lacks precision in description of sampling for isotopes, position of sampling for DIC (surface or deep), arguments for endmembers determination which is crucial in estuaries with River plumes. It thus requires a deep reworking before it can be published.

Main comments:

MC1- Typhoon influence: after presenting the study period and the occurrence of a typhoon "Rammasun", the authors write on page 4, line 17-18 "... that this study represented a typical situation of the area in terms of terrestrial material discharge". Yet, the typhoon brought heavy rain (and I suppose waves) which increased the Pearl River discharge to 26000 m3/s (double of the monthly average in July of 15000 m3/s). Later in the paper (page 8 line 8-18), the authors describe the changes of the bottom water composition at one station (A10, Fig. 5) which clearly show the changes in DIC, DIP, O2 and TSM concentration after the typhoon and until the end of the cruise. I think that the authors should reconsider the "typhoon" issue by saying that i) it has modified the system; ii) the system has restratified quickly due to large freshwater discharge iii) isotopic composition and DIC concentration before and long after the typhoon (1 week) may reflect the mineralization of OM (to be justified).

MC2- Calculation of the proportion of terrestrial versus marine OM mineralization in bottom water DIC: these calculations are made using a mixing model which is fine to me and well handled. However, due to the difficulty in defining the endmembers (which is common in such models), I believe that the uncertainty on the final proportion (75%

marine-25% terrestrial) is much larger than that reported. Furthermore, the arguments given for the endmember isotopic values are very weak: page 10 line 21-27 for 13C-TerrPOC, and page 11 line 1-7 for 13C-MarinePOC. In this last case, the authors quote 4 papers with values between -21.2 and -20.5 permil and finally choose -19.4+-0.8 which is out of the previous range. It is known that defining this value in estuaries or near estuaries is very complex and cannot be done with a restricted number of data (Harmelin-Vivien et al., 2008, CSR. Yet, changing this value from -19.4 permil to -21 permil would decrease de proportion of Terrestrial OC from 25% to 10%. With the uncertainty on the predicted values used in the equation and the uncertainty on the endmembers, the final uncertainty is certainly much larger than the 10% reported. The authors should give better arguments for their choice of isotopic composition of the end members and provide a sound estimation of the error propagation throughout their mixing model. They should also revise their estimation of terrestrial fraction, and modulate the conclusions.

MC3- Lack of precision for isotopic data: In stratified estuaries, the positioning of the sampling in the water column is crucial as strong vertical gradients (Fig. 5) are common in estuaries. Yet, when describing the isotopic measurements (page 5, line 1), it is not clear which samples were analysed. Were all samples analyzed as for DIC and O2? If so, which sample values are reported in Fig. 4 for 13C-DIC and which are reported in Fig 6d and Fig. 7? How were they chosen? Are there only 9 samples of subsurface waters for 13C-DIC measurements (reported on Fig. 7)? Clearly a better description of these data would be welcome with more maps (surface-bottom). See detailed comments below.

Detailed comments:

Abstract: English should be checked by a native speaker

Page 1- line 13: "differently sourced" should be changed to "different sources of"

Page 1- line 19: "hypoxias" is not used in the oceanographic literature. Use "hypoxic

events" or "hypoxic zones" or just "hypoxia"

Page 1- line 25: replace "marine sourced" by "from marine origin"

Page 1- line 26: replace "terrestrially sourced" by "from the continent"

Page 1- line 26:"eutrophication-stimulated marine sourced organic matter prevailed the oxygen consumption". Do the authors mean "marine organic matter stimulated by eutrophication dominated in the oxygen consumption"?

1. Introduction:

Page 2, line 5: Diaz and Rosenberg (2008) should be cited here. They showed in this paper that hypoxia was growing worldwide.

Page 2, line 14:"restoration" rather than "restoring"

2. Materials and Methods

Page 4, line 9: Fig. 1 is too small. You should mix leg 1 and leg 2 on the same map by superimposing crosses and circles, reduce the map of China and show the important stations (A8-A10). Indicate also Lindingyang Bay which is quoted in text

Page 4, line 17-18: "suggesting that this study represented a typical situation of the area in terms of terrestrial material discharge". This part of the sentence should be removed, and be consistent with paragraph 3.4 (Reinstatement of hypoxic stations. . .). See major comment MC1

Page 5, line 1-4: which samples were measured for 13C-DIC? Please specify (See MC3)

Page 5, line 16-26: The mixing model is described here, but not the final equations for DIC and 13C which are reported on page 10. I think that they should be all reported here for consistency.

3. Results

Page 6, line 7-8: rephrase "we noted that the survey . . .". No explanation is given why only the outside was covered during that first leg.

Page 6, line 10: Fig. 2 is much too small. There are too many data and type of data on this graph. You can either remove some data, cut bottom and surface in 2 different figures, or shift DIC in another graph. Two important things: the need to add 13C-DIC as it is the heart of this paper, use similar scales for all graphs of one type (e.g. O2) as this will allow easier and better reading of the graphs.

Page 6, line 14: "hypoxia lay more landward", I would say "central" more than landward.

Page 6, line 16: "the bloom zone was more westward" I see it more "eastward" (to the right on the map)

Page 6, line 19: "hypoxic zone was discovered southeast of Wanshan Islands" I see it more "southwest of the islands" (left side on map). Again, it is hard to see as the graphs are very small!

Page 6-line 21-22: "Hypoxia covered at least 800-900km2". Do the authors refer to stations with <2mg/l O2? They should provide the number of stations with DO<62 $\mu$mol/l (=2mg/l)

Page 7, line 12: Fig. 4 does not specify if the 13C-DIC is measured in surface or bottom water

Page 7, line 19: replace "d13C through" by "large d13C decrease"

Page 7, line 19: remove "geologically", replace by "geographically" or by "this station"

Page 7, line 21: Are sediments d13C as low as -35 permil? I do not see where this value can arise from.

Page 7, line 24: LT means Local time? If so please notify

Page 8, line 6: "might be the trail". Does it mean "might reflect"?

[Figure]

4. Discussion

Page 8, line 25: "when we chose S=34.6". In the Guo and Wong article, there are two depths of the profile which correspond to S=34.6: at 100 meters depth and at >1500 m. I suppose that the authors chose 100 meters which correspond to 2023 uM but they should specify.

Page 8, line 26: "DIC of 2023 umol/kg" cite the ref, I think it is Guo and Wong (2015)

Page 9, line 9-15: the authors explain well how they chose DIC endmember for river water/plume at S=10.8. But the way that 13C-DIC was calculated is not clear, it is all reported in the footnotes of Table 1. They should be exposed here more clearly, and the same approach should be taken for 13C as for DIC, which is not the case. Indeed, the 13C-plume endmember is calculated via simple mixing between freshwater and surface offshore water (footnote b of Table 1) which is not correct if biological uptake is active and differentiate between 12C and 13C. Please specify in text how the endmembers were calculated and justify your choices.

Page 9, line 18-21: "we chose Ca2+ as a conservative tracer...". Again, the endmembers for this element are not described in text, but in table 1 (footnote c). It is said that Ca2+ values of the endmembers were calculated by correlation with Salinity. So, I wonder if the correlation between prediction and observation for Ca2+ of Fig 6b is a test of the accuracy of the mixing model, because the mixing model is calibrated by T and S (line 16) and mostly S for surface waters, and Ca2+ is also calculated from S... Furthermore, it is not specified if the data points for Ca2+ refer to surface or subsurface water. Specify and remove argument if circular.

Page 10, line 21 to page 11 line 7: see my main comment (MC2) above on endmembers and uncertainty. The authors should justify their choice better, when the range is large as for terrestrial OM (-28.7 to -24.9 permil) and the choice is made to stick to one end of the range based on a few measurements. The situation is even worse for the offshore surface water POC where the chosen value (-19.4 permil) is out of the

range reported by the authors (see above). Furthermore, it is very difficult to assess pure plankton signature by surface water sampling as POM in estuaries or offshore is a mixture of marine POM and terrestrial POM (Harmelin-Vivien et al. 2008, CSR). The choice of this endmember is thus questionable and should be better justified.

Page 12: one possibility for the difference between the Changjiang and the Pear River could also be the presence of the megacity of Guangzhou and its fresh anthropic OM discharge into the river. As the typhoon washed away some of the material deposited in the river conduit, it could have led to more labile matter in the estuary.

5. Conclusion

Page 13, line 8-12: the authors should report a possible range for the share of terrestrial OM mineralization based on uncertainties in endmembers (between 10-25%) instead of the values reported here 27%.

Figures and Tables (see also comments in text above):

Fig. 6: The calculation of the "biological effect" on DIC and 13C-DIC is based on subsurface values. These values should be visible (dark dots?) on graph to identify the data used for deriving the signature of mineralized OM (on Fig. 7)

Fig. 8: The "pseudo-equations" on top and bottom HNO3+DIC→OM+O2 and OM+O2 →HNO3+DIC should be removed as they are not consistent nor balanced, and replaced by Primary Production (on top) and oxic mineralization (at the bottom). The graph inserted in the Figure is not readable, please consider removing.

Table 1: most text in notes should go in text as explanation and justification of endmember calculation.

---

## Referee Comment (RC2) · Anonymous Referee #2 · 29 Mar 2017

This paper determines the origin of the oxygen-consuming organic matter in the hypoxic zone of Pearl River Estuary, China. The approach is the same as that used for the Changjiang Estuary (ES&T 2016), utilizing C-13 value for the increased DIC in the hypoxic zone. I think that the approach is technically valid and provides useful information regarding the cause of hypoxia so as to make right policies for hypoxia remediation. However, authors do not show any advances in data analyses and interpretation, compared with the previous ES&T paper. Authors found that 73% of the oxygen-consuming organic matter is of marine source (the rest portion is terrestrial), which is different from the hypoxia in the Changjiang Estuary (100% marine organic matter contributions). Although authors do suggest some speculations for the cause of this difference, they do not go through thorough data analyses and interpretations about this issue. I suggest following trials to better interpret the data.

(1) Authors may compare all C-13 data for POC and DIC between the two estuaries together in the same plots, so that any systematic difference between the two estuaries can be examined.

(2) When I look at both data sets from the two estuaries, the difference in the contribution of marine organic matter for the two estuaries is within the uncertainties of this approach. Depending on how to omit the outlier in the relationship (Fig. 7 vs. Fig. 6 of ES&T), the proportion of marine organic matter varies significantly. The ES&T paper shows a large scattering for the slope (if hypoxic zone is collected separately) but reduced the error by including the subsurface layer data which is not reasonable. Therefore, I am not convinced with the argument that the sources of organic matter in the hypoxic zone of the two estuaries are different.

(3) I think that most of the oxidation happens in the surface sediment layer. In order to determine the reason for the difference between the two esturaries, authors should show surface sediment C-13 data (any difference between the two estuaries?). Otherwise present any difference in the characteristics of organic matter (C3/C4 plants vs. marine OM) in surface sediments.

(4) Authors state that "We chose the concentrations of Ca2+ as a conservative tracer to validate our model prediction, and the model values were in good accordance with the field-observed values..". In order to choose a conservative tracer, authors may use another conservative element instead of Ca which is not necessarily conservative in the coastal system. I think that Ca should be used to examine the effect of CaCO3 in this system, proving that no addition or removal of DIC associated with CaCO3. This result suggests that the DIC change in the hypoxia is solely owing to the organic matter dissolution.

(5) Although the contribution is relatively small, authors should account for the contribution of DOC in this system.

Minor comments

- Authors do not use correct significant figures for DO and DIC (i.e., 153.1 umol/kg for DO and 1900.7 umol/kg for DIC). Otherwise, do you measure such accurate numbers?

---

## Author Comment (AC1) · 30 May 2017

*Interactive Comment* on "Tracing the origin of the oxygen-consuming organic matter in the hypoxic zone in a large eutrophic estuary: the lower reach of the Pearl River Estuary, China" by Su et al.

Jianzhong Su[1], Minhan Dai[1*], Biyan He[1, 2], Lifang Wang[1], Jianping Gan[3], Xianghui Guo[1], Huade Zhao[1] and Fengling Yu[1]

[1]State Key Laboratory of Marine Environmental Science, Xiamen University, Xiamen, China
[2]College of Food and Biological Engineering, Jimei University, Xiamen, China
[3]Department of Mathematics and Division of Environment, Hong Kong University of Science and Technology, Kowloon, Hong Kong, China

*Correspondence to*: Minhan Dai (mdai@xmu.edu.cn)

**Anonymous Referee #1**

The paper by Su et al. deals with hypoxia and its causes in the Pearl River Estuary (PRE). Hypoxic events are a growing concern from human societies as they threat the environment and the resources on which coastal population lives. Furthermore, it strongly impacts the environment with a resilience time which is now largely unknown. Attribution of hypoxia to a type of organic matter mineralization has been seldom done, and is very interesting from a watershed manager point of view. It is the kind of effort that Su and his/her colleague have started, and for that reason their paper is of potential great interest. Indeed, they use DIC stable isotopic composition and a wealth of other data collected from two cruises in the Pearl River estuary and the adjacent coastal zone to estimate the contribution of terrestrial and marine organic matter to the decrease of oxygen in stratified bottom waters. The contribution is thus original as only a few attempts exist to quantify the source, e.g. in the South China Sea near Changjiang River, but the reading of the paper raises some fundamental issues which are poorly answered in the manuscript: was the perturbation of the typhoon small enough that it can be neglected (see page 4, line 17-18)? Is the proportion of 75-25% a robust estimate of the contribution of the marine and terrestrial pools? Can we quantify better the uncertainty? Overall the paper lacks precision in description of sampling for isotopes, position of sampling for DIC (surface or deep), arguments for endmembers determination which is crucial in estuaries with River plumes. It thus requires a deep reworking before it can be published.

[Response]: We are grateful that the reviewer valued our study. We also appreciate the critical and constructive comments, which have been fully considered in our revised MS. We will address these major concerns of the reviewer in our responses below.

**Main comments:**

MC1- Typhoon influence: after presenting the study period and the occurrence of a typhoon "Rammasun", the authors write on page 4, line 17-18 ". . . that this study represented a typical situation of the area in terms of terrestrial material discharge". Yet, the typhoon brought heavy rain (and I suppose waves) which increased the Pearl River discharge to 26000 m3/s (double of the monthly average in July of 15000 m3/s).

Later in the paper (page 8 line 8-18), the authors describe the changes of the bottom water composition at one station (A10, Fig. 5) which clearly show the changes in DIC, DIP, O2 and TSM concentration after the typhoon and until the end of the cruise. I think that the authors should reconsider the "typhoon" issue by saying that i) it has modified the system; ii) the system has restratified quickly due to large freshwater discharge iii) isotopic composition and DIC concentration before and long after the typhoon (1 week) may reflect the mineralization of OM (to be justified).

[Response]: The reviewer is correct. Typhoon "Rammasun" did increase riverine discharge and de-stratify the water column, which was however quickly re-established (Fig. 6). Following these suggestions from the reviewer, we have deleted the statement "that this study represented a typical situation of the area in terms of terrestrial material discharge". To further clarify the impact of the typhoon in terms of freshwater discharge, we have added daily river discharge measurements during 15-18 July, which were 19480, 26115, 22981 and 17540 $m^3$ $s^{-1}$, respectively. Note that our sampling was interrupted during 17-18 July due to the typhoon. Excluding the discharge rate during these two days, the average freshwater discharge rate during our actual sampling period was 16369 $m^3$ $s^{-1}$, which was slightly higher than the multi-year (2000-2011) monthly mean (15671 $m^3$ $s^{-1}$). Please see P4 L4-7 and P5 L19-21.

The impact of the typhoon has been illustrated in Fig. 6 and presented in Sect. 3.4. Unfortunately, we only sampled $\delta^{13}C_{DIC}$ at Station A10 on 20 July, which does not allow us to compare $\delta^{13}C_{DIC}$ values before and after the typhoon. However, other parameters we measured, such as DIC, $O_2$ and DIP, accurately reflect the mineralization of OM. Related revisions have been made in the revised MS in P5 L21-23.

[Figure]

**Figure 6.** Profiles of (a) temperature, (b) salinity, (c) DO, (d) DIC, (e) DIP, (f) TSM and their evolution during repeated sampling at Station A10.

MC2- Calculation of the proportion of terrestrial versus marine OM mineralization in bottom water DIC: these calculations are made using a mixing model which is fine to me and well handled. However, due to

the difficulty in defining the endmembers (which is common in such models), I believe that the uncertainty on the final proportion (75% marine-25% terrestrial) is much larger than that reported. Furthermore, the arguments given for the endmember isotopic values are very weak: page 10 line 21-27 for 13CTerrPOC, and page 11 line 1-7 for 13C-MarinePOC. In this last case, the authors quote 4 papers with values between -21.2 and -20.5 permil and finally choose -19.4+-0.8 which is out of the previous range. It is known that defining this value in estuaries or near estuaries is very complex and cannot be done with a restricted number of data (Harmelin-Vivien et al., 2008, CSR. Yet, changing this value from -19.4 permil to -21 permil would decrease de proportion of Terrestrial OC from 25% to 10%. With the uncertainty on the predicted values used in the equation and the uncertainty on the endmembers, the final uncertainty is certainly much larger than the 10% reported. The authors should give better arguments for their choice of isotopic composition of the end members and provide a sound estimation of the error propagation throughout their mixing model. They should also revise their estimation of terrestrial fraction, and modulate the conclusions.

[Response]: Again, we appreciate the reviewer's comments, which are indeed critical to this study. We have thus made great efforts to go through every detail regarding the selection of end members and the estimation of uncertainty.

1. Selection of $\delta^{13}C_{terr}$ and $\delta^{13}C_{mar}$

We initially defined the end-member values of $\delta^{13}C$ by using a composite value of particulate organic carbon (POC) and surface sediment organic carbon, which was not strictly correct because sedimentary organic matter often contains larger terrestrial components while POC is composed of more freshly produced organic matter (Middelburg and Nieuwenhuize, 1998). In the revised MS, we adopted the end-member values of $\delta^{13}C$ from POC.

Here, for the terrestrial end-member ($\delta^{13}C_{terr}$), we adopted the average $\delta^{13}C$ value of POC sampled near the Humen Outlet (S<4), which represents the predominant source of riverine material entering the estuary (He et al., 2010b). The mean POC value, -28.3±0.7 ‰ (n=7), is very similar to the freshwater $\delta^{13}C_{POC}$ value of -28.7 ‰ reported by Yu et al. (2010), which reflected a terrigenous mixture of the C3 plant fragments and forest soils. For the marine end-member ($\delta^{13}C_{mar}$), we calculated the mean surface water $\delta^{13}C_{POC}$ value (-19.4±0.8 ‰, n=8) from stations with S > 26 where significant phytoplankton blooms were observed, as indicated by DO supersaturation (DO% > 125 %) and relatively high pH values (> 8.3) and POC contents (5.3±2.4 %). This value is similar, although slightly heavier than the marine end-member used by Chen et al. (2008), who measured a $\delta^{13}C$ value of -20.9 ‰ in tow-net phytoplankton samples from outer Lingdingyang Bay, in the same region as this study. Additionally, He et al. (2010a) reported a $\delta^{13}C$ value of -20.8±0.4 ‰ in phytoplankton collected from the northern South China Sea (He et al., 2010a). These values are consistent enough for us to compile and obtain use average $\delta^{13}C_{mar}$ value of -20.5±0.9 ‰. This value also agrees very well with the reported stable carbon isotopic signature of marine organic matter in other coastal regions. For example, mean isotopic values for phytoplankton were reported as -20.3±0.6 ‰ in

Narragansett Bay (Gearing et al., 1984), -20.3±0.9 ‰ in Auke Bay and Fritz Cove (Goering et al., 1990), and -20.1±0.8 ‰ in the Gulf of Lions (Harmelin-Vivien et al., 2008).

2. Error propagation (uncertainties)

As shown in the revised Fig. 8, the composite uncertainty for the average $\delta^{13}C_{OCx}$ value is 1.1 ‰, which is close to the uncertainty (1.0 ‰) reported by Wang et al. (2016).

The fractional contribution of $OC_{terr}$ and $OC_{mar}$ to $\delta^{13}C_{OCx}$ is determined by $\delta^{13}C_{terr}$, $\delta^{13}C_{mar}$ and $\delta^{13}C_{OCx}$ values, and are calculated using the following equation:

$$f(\%)=\frac{\delta^{13}C_{mar}-\delta^{13}C_{OCx}}{\delta^{13}C_{mar}-\delta^{13}C_{terr}}\times100 \tag{9}$$

The composite uncertainty associated could be calculated by the following equation (Taylor, 1997; Han et al., 2012):

$$\varepsilon_{f(\%)}=\sqrt{\left(\frac{\partial(f)}{\partial(\delta^{13}C_{mar})}\times\delta([\delta^{13}C_{mar}])\right)^2+\left(\frac{\partial(f)}{\partial(\delta^{13}C_{terr})}\times\delta([\delta^{13}C_{terr}])\right)^2+\left(\frac{\partial(f)}{\partial(\delta^{13}C_{OCx})}\times\delta([\delta^{13}C_{OCx}])\right)^2}\times100$$

Using end-member values of -28.3±0.7 ‰ and -20.5±0.9 ‰ for $\delta^{13}C_{terr}$ and $\delta^{13}C_{mar}$, respectively, and the derived $\delta^{13}C_{OCx}$ value of -23.2±1.1 ‰, as justified above, the fractional contribution of marine organic matter is 65±16 %, while the terrestrial organic matter accounted for the remaining 35±16 %.

We have revised the MS accordingly as in P10 L21-22, P11 L4-26 and P13 L20-24.

[Figure]

**Figure 8.** $\Delta$ ($\delta^{13}C_{DIC}\times DIC$) vs. $\Delta DIC$ in the PRE. Samples were collected from subsurface water (> 5 m). The grey circles represent samples with AOU > 100 μmol kg⁻¹. $\Delta$ is the difference between the fieldobserved and model-predicted values. Also shown is data from the CJE reported by Wang et al. (2016). The straight and dashed lines indicate linear regression lines of data from the PRE and CJE, respectively.

MC3- Lack of precision for isotopic data: In stratified estuaries, the positioning of the sampling in the water column is crucial as strong vertical gradients (Fig. 5) are common in estuaries. Yet, when describing the isotopic measurements (page 5, line 1), it is not clear which samples were analysed. Were all samples analyzed as for DIC and O2? If so, which sample values are reported in Fig. 4 for 13C-DIC and which are reported in Fig 6d and Fig. 7? How were they chosen? Are there only 9 samples of subsurface waters for 13C-DIC measurements (reported on Fig. 7)? Clearly a better description of these data would be welcome with more maps (surface-bottom).

[Response]: We appreciate the comments. We measured DIC and $O_2$ at all stations where depth profiles were sampled. We primarily measured $\delta^{13}C_{DIC}$ along Transect A and at depth in the low oxygen layers (28 stations and 84 layers in total), which are now shown in revised Figs. 2i (surface in Leg 1), 2l (surface in Leg 2), 3i (bottom in Leg 1), and 3l (bottom in Leg 2). The sampling locations where $\delta^{13}C_{DIC}$ was measured along Transect A are shown in the revised Fig. 4f for Leg 1 and Fig. 4l for Leg 2. Figure 5 (original Fig. 4) now shows all the $\delta^{13}C_{DIC}$ (n=84) data collected in July 2014 with the surface and subsurface data distinguished.

Following the reviewer's suggestion, Fig. 7d (original Fig. 6d) has now been revised and includes all the subsurface data used in our three end-member mixing model (i.e. points within the solid triangle in Fig. 7a). In Fig. 8 (original Fig. 7), the data used for the linear regressions includes all the points in the revised Fig. 7d.

[Figure]

**Figure 2.** Surface water distribution of temperature, salinity, DO, Chl-a, DIC and $\delta^{13}C_{DIC}$ during Leg 1 (a–c, g–i) and Leg 2 (d–f, j–l).

[Figure]

**Figure 3.** Bottom water distribution of temperature, salinity, DO, Chl-a, DIC and $\delta^{13}C_{DIC}$ during Leg 1 (a–c, g–i) and Leg 2 (d–f, j–l). Note that the black lines in (c) and (f) indicate DO contours of 63 μM and 95 μM.

[Figure]

**Figure 4.** Profiles of temperature, salinity, DO, Chl-a, DIC and $\delta^{13}C_{DIC}$ along Transect A during Leg 1 (a–f) and Leg 2 (g–l). Note that the black lines in (c) and (i) indicate DO contours of 63 μM and 95 μM.

[Figure]

**Figure 5.** Distribution of $\delta^{13}C_{DIC}$ and $\delta^{13}C_{POC}$ with respect to salinity in the PRE. The up-facing and down-facing triangles denote surface and subsurface $\delta^{13}C_{DIC}$ data, respectively, from July 2014, while the open circles represent $\delta^{13}C_{POC}$ values in surface water from July 2015. Additionally, the plus signs and crosses show the $\delta^{13}C_{DIC}$ and $\delta^{13}C_{POC}$ data, respectively, from the CJE in Wang et al. (2016).

[Figure]

**Figure 7.** (a) Potential temperature ($\theta$) (℃) vs. salinity in the PRE and adjacent coastal waters (open circles) based on data collected during the July 2014 cruise. The three end-members are shown as different coloured symbols. The blue triangles represent data collected during the August 2011 cruise in the CJE (Wang et al., 2016); (b) Correlation between the field-observed $Ca^{2+}$ ($Ca^{2+}_{obs}$) and model-predicted $Ca^{2+}$ ($Ca^{2+}_{pre}$). The straight line denotes a linear regression line of both surface (square) and subsurface (diamond) data; (c), (d) Relationship between observed and model-predicted DIC and $\delta^{13}C_{DIC}$ values. The straight line represents a 1:1 reference line. Note that the grey dots in Fig. 7c identify data also in Fig. 7d; and (e) Correlation of $\Delta$DIC vs. AOU for all subsurface water data. $\Delta$DIC is the difference between the field-observed and model-predicted DIC concentrations. Also shown is the data from Wang et al. (2016). The straight and dashed lines indicate linear regressions of data from the PRE and CJE, respectively.

**Detailed comments:**

Abstract: English should be checked by a native speaker

[Response]: We have had our revised MS proofread again by a native English speaker.

Page 1- line 13: "differently sourced" should be changed to "different sources of"

[Response]: Accepted. We have changed "differently sourced" to "different sources of". Please see P1 L13.

Page 1- line 19: "hypoxias" is not used in the oceanographic literature. Use "hypoxic events" or "hypoxic zones" or just "hypoxia"

[Response]: Accepted. We have corrected "hypoxias" to "hypoxia" or "hypoxic zones" throughout the revised MS. Please see P1 L19.

Page 1- line 25: replace "marine sourced" by "from marine origin"

[Response]: Accepted. We have replaced "marine sourced" with "from marine origin" throughout the revised MS where appropriate. However, on P1 L25 we have changed this to "derived from marine sources," which fit better with the revised text.

Page 1- line 26: replace "terrestrially sourced" by "from the continent"

[Response]: Accepted. We have replaced "terrestrially sourced" by "from the continent" or "terrestrial" throughout the revised MS. Please see P1 L26.

Page 1- line 26:"eutrophication-stimulated marine sourced organic matter prevailed the oxygen consumption". Do the authors mean "marine organic matter stimulated by eutrophication dominated in the oxygen consumption"?

[Response]: The reviewer is right, and we have changed "eutrophication-stimulated marine sourced organic matter prevailed the oxygen consumption" to "marine organic matter stimulated by eutrophication dominated oxygen consumption". Please see P1 L27-28.

1. Introduction:

Page 2, line 5: Diaz and Rosenberg (2008) should be cited here. They showed in this paper that hypoxia was growing worldwide.

[Response]: Accepted. We have cited Diaz and Rosenberg (2008) here. Please see P2 L2-3.

Page 2, line 14:"restoration" rather than "restoring"

[Response]: This text has been significantly changed in the revised MS and the word "restoration" has been removed completely. Please see P2 L10-13.

2. Materials and Methods

Page 4, line 9: Fig. 1 is too small. You should mix leg 1 and leg 2 on the same map by superimposing crosses and circles, reduce the map of China and show the important stations (A8-A10). Indicate also Lindingyang Bay which is quoted in text

[Response]: Accepted. We have put both Legs 1 and 2 on one figure and inserted an inset map of China. We have marked Lingdingyang Bay in the revised Fig. 1.

[Figure]

**Figure 1.** Map of the Pearl River Estuary and adjacent coastal waters. The open circles denote the Leg 1 stations on 13–16 July 2014 and the crosses represent the Leg 2 stations on 19–27 July 2014. Note that the filled diamond is the location of Station A10.

Page 4, line 17-18: "suggesting that this study represented a typical situation of the area in terms of terrestrial material discharge". This part of the sentence should be removed, and be consistent with paragraph 3.4 (Reinstatement of hypoxic stations. . .). See major comment MC1

[Response]: Accepted and we have removed this sentence. Please also see our responses to MC1.

Page 5, line 1-4: which samples were measured for 13C-DIC? Please specify (See MC3)

[Response]: Please see our response to MC3.

Page 5, line 16-26: The mixing model is described here, but not the final equations for DIC and 13C which are reported on page 10. I think that they should be all reported here for consistency.

[Response]: Accepted. We have added Eq. (4) and (5) to define the predicted values of DIC and $\delta^{13}C_{DIC}$. Please see P5 L13-14.

$$DIC_{RI} \times F_{RI} + DIC_{SW} \times F_{SW} + DIC_{SUB} \times F_{SUB} = DIC_{pre} \tag{4}$$

$$\frac{\delta^{13}C_{DICRI} \times DIC_{RI} \times F_{RI} + \delta^{13}C_{DICSW} \times DIC_{SW} \times F_{SW} + \delta^{13}C_{DICSUB} \times DIC_{SUB} \times F_{SUB}}{DIC_{pre}} = \delta^{13}C_{DICpre} \tag{5}$$

3. Results

Page 6, line 7-8: rephrase "we noted that the survey . . .". No explanation is given why only the outside was covered during that first leg.

[Response]: Our sampling was interrupted during 17-18 July due to the typhoon. Thus, we only covered outer Lingdingyang Bay during Leg 1. We have rephrased "The interruption of Leg 1 due to the typhoon (July 17-18) led to a smaller survey area, covering only outside Lingdingyang Bay (traditionally regarded as the PRE), while Leg 2 covered Lingdingyang Bay from the Humen Outlet to the adjacent coastal sea". Please see P5 L23-26.

Page 6, line 10: Fig. 2 is much too small. There are too many data and type of data on this graph. You can either remove some data, cut bottom and surface in 2 different figures, or shift DIC in another graph. Two important things: the need to add 13C-DIC as it is the heart of this paper, use similar scales for all graphs of one type (e.g. O2) as this will allow easier and better reading of the graphs.

[Response]: Per the reviewer's suggestion, we have split the surface and bottom water data into two figures (Figs. 2 & 3). Also, we added $\delta^{13}C_{DIC}$ plots in both figures. We are now using similar scales for both surface and bottom data. Moreover, we added $\delta^{13}C_{DIC}$ plots and used similar scales for the profiles along Transect A in the revised Fig. 4. The related descriptions have been revised accordingly. Please see our response to MC3.

Page 6, line 14: "hypoxia lay more landward", I would say "central" more than landward.

[Response]: Accepted. We have replaced "lay more landward" by "was located more centrally". Please see P6 L11.

Page 6, line 16: "the bloom zone was more westward" I see it more "eastward" (to the right on the map)

[Response]: By comparing the surface distributions of DO, Chl-a and pH during Leg 2, we have revised this to say "During Leg 2, there were three patches of high productivity, south of Huangmaohai, at the PRE entrance, and off Hong Kong. The central region of high productivity had the highest DO%, greater than

140% at Station A14, and was characterized by relatively high concentrations of Chl-a (7.8 µg kg$^{-1}$) and low concentrations of DIC (1737.3 µmol kg$^{-1}$)." Please see P6 L6-12.

[Figure]

Figure S1. The surface distribution of pH during Leg 2.

Page 6, line 19: "hypoxic zone was discovered southeast of Wanshan Islands" I see it more "southwest of the islands" (left side on map). Again, it is hard to see as the graphs are very small!

[Response]: The reviewer is right. We have corrected it as "southwest of the Wanshan Islands". Please see P6 L16-17.

Page 6-line 21-22: "Hypoxia covered at least 800-900km2". Do the authors refer to stations with <2mg/l O2? They should provide the number of stations with DO<62 µmol/l (=2mg/l)

[Response]: Accepted. We have revised our estimate of the surface area of the hypoxic zone as well as the text. The section now reads "…our results suggest it covered an area of > 280 km$^2$ during Leg 1 and > 290 km$^2$ during Leg 2 according to the definition of hypoxia as DO < 2 mg L$^{-1}$ or 63 µM, or an area of > 900 km$^2$ during Leg 1 and > 800 km$^2$ during Leg 2 assuming the threshold of the oxygen-deficit zone was < 3 mg L$^{-1}$ or 95 µM (Rabalais et al., 2010; Zhao et al., 2017).". We have also added 63 µM and 95 µM DO contours to Figs. 3c, 3f, 4c and 4i. Please see P6 L20-24.

Page 7, line 12: Fig. 4 does not specify if the 13C-DIC is measured in surface or bottom Water

[Response]: We now distinguish between surface and subsurface $\delta^{13}C_{DIC}$ data in the revised Fig. 5. Please see our response to MC3.

Page 7, line 19: replace "d13C through" by "large d13C decrease"

[Response]: Accepted. We have replaced "$\delta^{13}C_{POC}$ trough" by "large $\delta^{13}C_{POC}$ decrease". Please see P7 L18

Page 7, line 19: remove "geologically", replace by "geographically" or by "this station"

[Response]: Accepted. We have replaced "Geologically" by "Geographically". Please see P7 L19.

Page 7, line 21: Are sediments d13C as low as -35 permil? I do not see where this value can arise from.

[Response]: We are grateful that the reviewer spotted this specific data point. By examining the previous studies in the region (e.g., Hu et al., 2005; Ye et al., 2010; He et al., 2010), we judged that this data point may be erroneous and thus have deleted it in the revised MS. Please see revised Fig. 5 in the response to MC3.

Page 7, line 24: LT means Local time? If so please notify

[Response]: Accepted. We have specified LT as Local Time. Please see P7 L24.

Page 8, line 6: "might be the trail". Does it mean "might reflect"?

[Response]: Accepted. We have revised "might be the trail" to "might reflect". Please see P8 L2.

4. Discussion

Page 8, line 25: "when we chose S=34.6". In the Guo and Wong article, there are two depths of the profile which correspond to S=34.6: at 100 meters depth and at >1500 m. I suppose that the authors chose 100 meters which correspond to 2023 uM but they should specify.

[Response]: The reviewer is right. We have revised this to read "Here, by choosing S=34.6 as the offshore subsurface water salinity end-member, we obtained a DIC value of ~2023 $\mu mol\ kg^{-1}$, similar to the value at ~100 m depth adopted by Guo and Wong (2015)." Please see P8 L22-25.

Page 8, line 26: "DIC of 2023 umol/kg" cite the ref, I think it is Guo and Wong (2015)

[Response]: Thanks for the comment. Yes, it is Guo and Wong (2015).

Page 9, line 9-15: the authors explain well how they chose DIC endmember for river water/plume at S=10.8. But the way that 13C-DIC was calculated is not clear, it is all reported in the footnotes of Table 1. They should be exposed here more clearly, and the same approach should be taken for 13C as for DIC, which is not the case. Indeed, the 13C-plume endmember is calculated via simple mixing between freshwater and surface offshore water (footnote b of Table 1) which is not correct if biological uptake is active and differentiate between 12C and 13C. Please specify in text how the endmembers were calculated and justify your choices.

[Response]: The reviewer is correct. Since biological alteration may differentiate between $^{12}C$ and $^{13}C$, we derived the plume-water $^{13}C_{DIC}$ end-member value based on the conservative mixing curve of freshwater and seawater end-members, where little biological alteration occurred. We have moved the footnotes into the main text with more explanations. "The $\delta^{13}C_{DIC}$ value was 0.6±0.2 ‰ in the offshore surface seawater at S=~33.7, where nutrient ($NO_3^-+NO_2^-$ and DIP) concentrations were close to their detection limits and DO was nearly saturated, indicating little biological activity. As DIC remained overall conservative when salinity was <10.8, the $\delta^{13}C_{DIC}$ value of -11.4±0.2 ‰ at S<0.4 is representative of the freshwater source. Assuming the plume water is a mixture of freshwater and offshore surface seawater, the initial plume endmember of $\delta^{13}C_{DIC}$ at S=10.8 can be calculated via an isotopic mass balance (-7.0±0.8 ‰)". Please see P9 L9-15.

Page 9, line 18-21: "we chose Ca2+ as a conservative tracer. . .". Again, the endmembers for this element are not described in text, but in table 1 (footnote c). It is said that Ca2+ values of the endmembers were calculated by correlation with Salinity. So, I wonder if the correlation between prediction and observation for Ca2+ of Fig 6b is a test of the accuracy of the mixing model, because the mixing model is calibrated by T and S (line 16) and mostly S for surface waters, and Ca2+ is also calculated from S. . . Furthermore, it is not specified if the data points for Ca2+ refer to surface or subsurface water. Specify and remove argument if circular.

[Response]: We apologize for the confusion. We only derived end-member values of the plume and surface seawater using the Ca-salinity relationship, and our $Ca^{2+}$ data collected during the 2014 cruise was only for the purpose of validating our three end-member mixing model. In the revised version, the $Ca^{2+}$ values of the plume and surface seawater end-member were derived independently from a conservative mixing calculation ($Ca^{2+}$ vs. S) based on a 3 years of surface data during the summer (August 2012, July 2014 and July 2015). As shown in Fig. S2, we obtained mean values of 3670±16 μmol kg$^{-1}$ at S=10.8 as the plume end-member and 9776±132 μmol kg$^{-1}$ at S=33.7 as the offshore surface seawater end-member. The subsurface $Ca^{2+}$ end-member was chosen from the measured value at S=~34.6 during the 2014 cruise. In the revised Fig. 7b (original Fig. 6d), all of the $Ca^{2+}$ data from both surface and subsurface waters (shown with distinguished symbols) are plotted to compare with model-predicted $Ca^{2+}$ values. These values agree well, which strongly supports our model predictions. Please see revised Fig. 7b in the response to MC3.

[Figure]

Figure S2. Historical surface $Ca^{2+}$ data plotted against salinity in the PRE during the summer.

Page 10, line 21 to page 11 line 7: see my main comment (MC2) above on endmembers and uncertainty. The authors should justify their choice better, when the range is large as for terrestrial OM (-28.7 to -24.9 permil) and the choice is made to stick to one end of the range based on a few measurements. The situation is even worse for the offshore surface water POC where the chosen value (-19.4 permil) is out of the range reported by the authors (see above). Furthermore, it is very difficult to assess pure plankton signature by surface water sampling as POM in estuaries or offshore is a mixture of marine POM and terrestrial POM (Harmelin-Vivien et al. 2008, CSR). The choice of this endmember is thus questionable and should be better justified.

[Response]: Please see our response to MC2 regarding the selection of $\delta^{13}C_{terr}$ and $\delta^{13}C_{mar}$ end-member values.

Page 12: one possibility for the difference between the Changjiang and the Pear River could also be the presence of the megacity of Guangzhou and its fresh anthropic OM discharge into the river. As the typhoon washed away some of the material deposited in the river conduit, it could have led to more labile matter in the estuary.

[Response]: We very much agree with this comment. We have added it to the discussion of the bioavailability of $OC_{terr}$ in Sect.4.3 as "Moreover, the increased precipitation and runoff during the typhoon may have mobilized additional fresh anthropogenic OM from surrounding megacities (e.g. Guangzhou, Shenzhen and Zhuhai) deposited in the river channel, which could lead to more labile $OC_{terr}$ in the PRE". Please see P13 L6-9.

5. Conclusion

Page 13, line 8-12: the authors should report a possible range for the share of terrestrial OM mineralization based on uncertainties in endmembers (between 10-25%) instead of the values reported here 27%.

[Response]: Accepted. We have revised it as "…we demonstrated that the organic matter decomposed via aerobic respiration in the stratified subsurface waters of the lower PRE and adjacent coastal waters was predominantly $OC_{mar}$ (49-81 %, mean 65 %), with a significant portion of $OC_{terr}$ also decomposed (19-51 %, mean 35 %)." Please see P13 L20-24.

Figures and Tables (see also comments in text above):

Fig. 6: The calculation of the "biological effect" on DIC and 13C-DIC is based on subsurface values. These values should be visible (dark dots?) on graph to identify the data used for deriving the signature of mineralized OM (on Fig. 7)

[Response]: Accepted. Please refer to our response to MC3. Also, we have identified the data in revised Figs. 7c (grey dots) that corresponds to the data in Fig. 7d.

Fig. 8: The "pseudo-equations" on top and bottom HNO3+DIC→OM+O2 and OM+O2 →HNO3+DIC should be removed as they are not consistent nor balanced, and replaced by Primary Production (on top) and oxic mineralization (at the bottom). The graph inserted in the Figure is not readable, please consider removing.

[Response]: Accepted. We have changed the previous equations to balanced equations of primary production and oxic mineralization. Also, the small inserted graph has been removed.

[Figure]

**Figure 9.** A conceptual diagram illustrating the source partitioning of oxygen-consuming organic matter ($OC_{mar}$ vs. $OC_{terr}$) within the hypoxic zone in the lower PRE and the adjacent coastal area. See Sect. 5 for explanations.

Table 1: most text in notes should go in text as explanation and justification of endmember calculation.

[Response]: Accepted. We have moved our notes on the selection of $\delta^{13}C_{DIC}$ values in the plume end-member into the text. Please see P22.

References:

Gearing, J. N., Gearing, P. J., Rudnick, D. T., Requejo, A. G., and Hutchins, M. J.: Isotopic variability of organic carbon in a phytoplankton-based, temperate estuary, Geochim. Cosmochim. Acta, 48, 1089-1098, doi:10.1016/0016-7037(84)90199-6, 1984.

Goering, J., Alexander, V., and Haubenstock, N.: Seasonal variability of stable carbon and nitrogen isotope ratios of organisms in a North Pacific Bay, Estuar. Coast. Shelf Sci., 30, 239-260, doi:10.1016/0272-7714(90)90050-2, 1990.

Han, A., Dai, M., Kao, S.-J., Gan, J., Li, Q., Wang, L., Zhai, W., and Wang, L.: Nutrient dynamics and biological consumption in a large continental shelf system under the influence of both a river plume and coastal upwelling, Limnol. Oceanogr., 57, 486-502, doi:10.4319/lo.2012.57.2.0486, 2012.

Harmelin-Vivien, M., Loizeau, V., Mellon, C., Beker, B., Arlhac, D., Bodiguel, X., Ferraton, F., Hermand, R., Philippon, X., and Salen-Picard, C.: Comparison of C and N stable isotope ratios between surface

particulate organic matter and microphytoplankton in the Gulf of Lions (NW Mediterranean), Cont. Shelf Res., 28, 1911-1919, 2008.

He, B., Dai, M., Huang, W., Liu, Q., Chen, H., and Xu, L.: Sources and accumulation of organic carbon in the Pearl River Estuary surface sediment as indicated by elemental, stable carbon isotopic, and carbohydrate compositions, Biogeosciences, 7, 3343-3362, doi:10.5194/bg-7-3343-2010, 2010a.

He, B., Dai, M., Zhai, W., Wang, L., Wang, K., Chen, J., Lin, J., Han, A., and Xu, Y.: Distribution, degradation and dynamics of dissolved organic carbon and its major compound classes in the Pearl River estuary, China, Mar. Chem., 119, 52-64, doi:10.1016/j.marchem.2009.12.006, 2010b.

Middelburg, J. J. and Nieuwenhuize, J.: Carbon and nitrogen stable isotopes in suspended matter and sediments from the Schelde Estuary, Mar. Chem., 60, 217-225, doi:10.1016/S0304-4203(97)00104-7, 1998.

Rabalais, N. N., Díaz, R. J., Levin, L. A., Turner, R. E., Gilbert, D., and Zhang, J.: Dynamics and distribution of natural and human-caused hypoxia, Biogeosciences, 7, 585-619, doi:10.5194/bg-7-585-2010, 2010.

Taylor, J. R.: Introduction to error analysis, the Study of Uncertainties in Physical Measurements, 2nd Edition, University Science Books, New York, 1997.

Zhao, H.-D., Kao, S.-J., Zhai, W.-D., Zang, K.-P., Zheng, N., Xu, X.-M., Huo, C., and Wang, J.-Y.: Effects of stratification, organic matter remineralization and bathymetry on summertime oxygen distribution in the Bohai Sea, China, Cont. Shelf Res., 134, 15-25, doi:10.1016/j.csr.2016.12.004, 2017.

---

## Author Comment (AC2) · 30 May 2017

*Interactive Comment* on "Tracing the origin of the oxygen-consuming organic matter in the hypoxic zone in a large eutrophic estuary: the lower reach of the Pearl River Estuary, China" by Su et al.

Jianzhong Su[1], Minhan Dai[1*], Biyan He[1, 2], Lifang Wang[1], Jianping Gan[3], Xianghui Guo[1], Huade Zhao[1] and Fengling Yu[1]

[1]State Key Laboratory of Marine Environmental Science, Xiamen University, Xiamen, China
[2]College of Food and Biological Engineering, Jimei University, Xiamen, China
[3]Department of Mathematics and Division of Environment, Hong Kong University of Science and Technology, Kowloon, Hong Kong, China

*Correspondence to*: Minhan Dai (mdai@xmu.edu.cn)

**Anonymous Referee #2**

This paper determines the origin of the oxygen-consuming organic matter in the hypoxic zone of Pearl River Estuary, China. The approach is the same as that used for the Changjiang Estuary (ES&T 2016), utilizing C-13 value for the increased DIC in the hypoxic zone. I think that the approach is technically valid and provides useful information regarding the cause of hypoxia so as to make right policies for hypoxia remediation. However, authors do not show any advances in data analyses and interpretation, compared with the previous ES&T paper. Authors found that 73% of the oxygen-consuming organic matter is of marine source (the rest portion is terrestrial), which is different from the hypoxia in the Changjiang Estuary (100% marine organic matter contributions). Although authors do suggest some speculations for the cause of this difference, they do not go through thorough data analyses and interpretations about this issue. I suggest following trials to better interpret the data.

(1) Authors may compare all C-13 data for POC and DIC between the two estuaries together in the same plots, so that any systematic difference between the two estuaries can be examined.

[Response]: Thanks for the constructive comment. We have added data from the CJE reported by Wang et al. (2016) and expanded the comparison between the two systems. We see similarities between these two systems: a) As shown in Fig. 5 (original Fig. 4), there is little difference between $\delta^{13}C_{DIC}$ and $\delta^{13}C_{POC}$ values in the marine end-member. b) In Fig. 7e, the amplitude of $\Delta DIC$ and AOU reveals a similar intensity of OM biodegradation, and the slope of $\Delta DIC$ vs. AOU ($0.71 \pm 0.03$ vs. $0.65 \pm 0.04$) indicates a predominance of aerobic respiration in the two systems. c) As seen from Table 2, there is no significant difference in $\delta^{13}C$ surface sediment values within the hypoxic zone between the PRE and CJE. We also note important differences between these two systems: a) In Fig. 5, the $\delta^{13}C_{DIC}$ and $\delta^{13}C_{POC}$ values of the freshwater end-members show some dissimilarity, with lighter values in the PRE (-11.4±0.2 ‰, -28.3±0.7 ‰) than in the CJE (-8.8 ‰, -24.4±0.2 ‰). b) Figure 7a indicates that the water temperature of the PRE is generally higher than in the CJE. For instance, the temperatures of surface and subsurface seawater endmembers in the PRE are 2-3 °C higher than in the CJE. c) From a spatial point of view, the distance from the river mouth to the hypoxic zone in the CJE is 2-3 times longer than in the PRE, possibly resulting in a longer travel time of $OC_{terr}$. Please see P12 L10-27.

**Table 2.** Comparison of $\delta^{13}C$ values in surface sediments within the hypoxic zone[a] between the PRE and CJE.

| $\delta^{13}C$ (‰) | Mean±SD | Stations involved | References |
|---|---|---|---|
| | | Pearl River Estuary | |
| -23.4 ~ -22.1 | -22.9±0.5 | A4, A5, C1-C4, D1 | Hu et al. 2006 |
| -23.2 ~ -22.3 | -22.7±0.5 | 28, 29, 30 | Zong et al. 2006 |
| -23.6 ~ -21.5 | -22.5±1.1 | E8-1, E7A, S7-1, S7-2 | He et al. 2010a |
| -[b] | -23.1±0.6 | Clustering groups G6 and G7 | Yu et al. 2010 |
| Average | -22.8±0.6 | | |
| | | Changjiang Estuary | |
| -22.9 ~ -20.9 | -21.8±0.6 | -[c] | Tan et al. 1991 |
| -22.4 ~ -19.9 | -21.2±1.0 | 32, 37, 38, 42, 48, 49, 54, 56, 64 | Kao et al. 2003 |
| -22.7 ~ -20.8 | -22.0±0.8 | H1-12, H2-10, H2-11, S1-2, S2-4 | Xing et al. 2011 |
| -23.5 ~ -20.4 | -22.6±1.0 | 3, 12, 13, 20-25 | Yao et al. 2014 |
| Average | -21.9±1.0 | | |

[a]In the PRE, the data is from similar sites to our present study, which is in the northeast (Leg 1) and southwest (Leg 2) of the Wanshan Islands. While in the CJE, the hypoxic zone is located around 30.0 °N–32.0 °N, 122.7 °E–123.2 °E, which is frequently reported in previous studies (Li et al., 2002; Zhu et al., 2011; Wang et al., 2016).

[b]The authors provide an average value of clustering groups instead of individual data from each site.

[c]In Fig. 7 of Tan et al. (1991), the sampling sites are shown without numbers.

(2) When I look at both data sets from the two estuaries, the difference in the contribution of marine organic matter for the two estuaries is within the uncertainties of this approach. Depending on how to omit the outlier in the relationship (Fig. 7 vs. Fig. 6 of ES&T), the proportion of marine organic matter varies significantly. The ES&T paper shows a large scattering for the slope (if hypoxic zone is collected separately) but reduced the error by including the subsurface layer data which is not reasonable. Therefore, I am not convinced with the argument that the sources of organic matter in the hypoxic zone of the two estuaries are different.

[Response]: We appreciate this critical comment from the reviewer.

a) After carefully re-examining all of the end-members and their error propagations, we have derived a $\delta^{13}C_{OCx}$ value of -23.2±1.1 ‰. For more details please see our response to MC2 from reviewer #1. This value of -23.2±1.1 ‰ is statistically different from the marine-derived OM, which has an isotopic composition of -20.5±0.9 ‰. Therefore, OM than that derived from marine sources must have been contributing to the DO consumption in the PRE hypoxic zone.

b) As clearly depicted in comment (1), the PRE has higher water temperatures, relatively lighter isotopic compositions of DIC and POC, and shorter travel times of $OC_{terr}$ than the CJE, which has implications for microbial activity and the bioavailability of $OC_{terr}$ and $OC_{mar}$. Combined with the dissimilarity of bacterial community structures in the PRE and CJE, the aforementioned factors can explain the observed differences in the relative distribution of terrestrial and marine OM contributing to oxygen consumption in these two systems. Please see our more detailed description in Sect. 4.3.

(3) I think that most of the oxidation happens in the surface sediment layer. In order to determine the reason for the difference between the two esturaries, authors should show surface sediment C-13 data (any difference between the two estuaries?). Otherwise present any difference in the characteristics of organic matter (C3/C4 plants vs. marine OM) in surface sediments.

[Response]: Although the sediment oxygen demand might be significant, the present study suggests that the oxidation happens mostly in the water column, which is quite similar to observations of CJE hypoxia. Jinwen Liu has demonstrated that the community respiration rate in the subsurface water of the CJE could nearly account for the entire decrease in oxygen after the passing of the typhoon (Liu, 2014). In addition, Wang et al. (2010) claims that water column respiration plays a greater role than sediment respiration in contributing to the establishment of hypoxia in the CJE, and that sinking POC derived from diatom blooms dominate this oxygen depletion in the water column (Wang et al., 2017). In both cases, the decline in oxygen happened after the typhoon passed and resuspended sediment concentrations in bottom waters returned to pre-typhoon levels (Wang et al., 2016). We cannot distinguish between the contribution of resuspended sediment vs. sinking POC to oxygen depletion, because both of these isotopic signals would have been reflected in oxidation product (DIC) and the derived $\delta^{13}C$ value of the remineralized OC ($\delta^{13}C_{OCx}$).

As seen from Table 2, there is no significant difference of $\delta^{13}C$ surface sediment values within the hypoxic zones of the PRE and CJE.

In the PRE, Yu et al. (2010) showed that the $\delta^{13}C$ value of surface sediments in the estuary increases from -25.0 ‰ at the freshwater-dominated reaches to -21.0 ‰ at the marine-dominated mouth, suggesting a weakening of terrestrial/freshwater (C3 plants detritus and soil) inputs and a strengthening of marine contribution to sediments seaward. Agricultural inputs (enhanced fertilization and an increase in the proportion of C4 plants such sugarcane) and anthropogenic inputs (crude oil and PAHs, PCBs and organochrolinate pesticides) would elevate the organic $\delta^{13}C$ of the sediment. In the CJE, Li et al. (2014) also concludes that the distribution pattern of $\delta^{13}C$ showed a general trend of enrichment seaward and

southward from the CJE. While the depleted $\delta^{13}C$ values were largely attributed to terrestrial inputs of lignin-rich C3 vascular plant debris coupled with coarse sediments, the enriched values were primarily thought to be derived from marine phytoplankton in the relict sands of deeper waters.

(4) Authors state that "We chose the concentrations of Ca2+ as a conservative tracer to validate our model prediction, and the model values were in good accordance with the field-observed values..". In order to choose a conservative tracer, authors may use another conservative element instead of Ca which is not necessarily conservative in the coastal system. I think that Ca should be used to examine the effect of CaCO3 in this system, proving that no addition or removal of DIC associated with CaCO3. This result suggests that the DIC change in the hypoxia is solely owing to the organic matter dissolution.

[Response]: It is true that other elements, like $\delta^{18}O$, can be used as alternative conservative tracers to observe the physical mixing of different water masses in coastal regions. However, in our case, $Ca^{2+}$ was a good choice as a conservative tracer given that $CaCO_3$ precipitation or dissolution was not significant, as judged by the strong linear relationship between surface water $Ca^{2+}$ and salinity (Fig. S2), and the oversaturation state of aragonite ($\Omega_{arag}$=2.6±0.7) in subsurface waters. We used a version of CO2Sys_2.1 to derive $\Omega_{arag}$ from DIC and TA. DIC data is shown in Fig. 7c. The constants K1, K2 from Cai and Wang (1998) and KHSO4 from Dickson (1990) are adopted in the $\Omega_{arag}$ calculations.

As the slope of $\Delta$DIC vs. AOU in the subsurface water was 0.71±0.03, approximating classic Redfield stoichiometry (i.e., 106/138=0.77), we concluded that no addition of DIC was attributed to $CaCO_3$, and the DIC change in hypoxic zone was due solely to organic matter remineralization.

[Figure]

Figure S2. Historical surface $Ca^{2+}$ data plotted against salinity in the PRE during the summer.

(5) Although the contribution is relatively small, authors should account for the contribution of DOC in this system.

[Response]: The reviewer brought up an important issue. Though it remains to be further explored, we contend that the contribution from DOC in consuming oxygen in the hypoxic zone should be minor as the reviewer expected. In the lower PRE (S>20), the DOC versus salinity showed an almost linear distribution, indicating conservative mixing and therefore minor oxidation of DOC (He et al., 2010). As shown in Wang et al. (2016), $\delta^{13}C_{DOC}$ is not generally significantly different from $\delta^{13}C_{POC}$ in most systems. Additionally, the oxidation of DOC in the lower PRE would only affect the estimates of non-conservative DIC mixing, rather than our estimates of the relative contributions of $OC_{terr}$ or $OC_{mar}$ to the oxygen-consuming OM pool. Thus, we did not account for the contribution of DOC to oxygen consumption.

**Minor comments**

- Authors do not use correct significant figures for DO and DIC (i.e., 153.1 umol/kg for DO and 1900.7 umol/kg for DIC). Otherwise, do you measure such accurate numbers?

[Response]: Thanks for the comment. The precision of DO measurement is ±1 μmol/kg. The precision is ± 2 μmol/kg for DIC measurements. We kept one decimal place for DO/DIC and use this data to plot the horizontal distributions in revised Figs. 2 & 3 and profiles along Transect A in revised Fig. 4.

References:

He, B., Dai, M., Zhai, W., Wang, L., Wang, K., Chen, J., Lin, J., Han, A., and Xu, Y.: Distribution, degradation and dynamics of dissolved organic carbon and its major compound classes in the Pearl River estuary, China, Mar. Chem., 119, 52-64, doi:10.1016/j.marchem.2009.12.006, 2010.
Li, D., Zhang, J., Huang, D., Wu, Y., and Liang, J.: Oxygen depletion off the Changjiang (Yangtze River) estuary, Sci. China Ser. D-Earth Sci., 45, 1137-1146, doi:10.1360/02yd9110, 2002.
Liu, J.: On the respiration and ocean acidification in the coastal ocean, Ph.D., College of the Environment & Ecology, Xiamen University, Xiamen, 178 pp., 2014.
Wang, B., Chen, J., Jin, H., Li, H., Huang, D., and Cai, W.-J.: Diatom bloom-derived bottom water hypoxia off the Changjiang estuary, with and without typhoon influence, Limnol. Oceanogr., doi:10.1002/lno.10517, 2017.
Wang, H., Dai, M., Liu, J., Kao, S.-J., Zhang, C., Cai, W.-J., Wang, G., Qian, W., Zhao, M., and Sun, Z.: Eutrophication-Driven Hypoxia in the East China Sea off the Changjiang Estuary, Environ. Sci. Technol., 50, 2255-2263, doi:10.1021/acs.est.5b06211, 2016.
Zhu, Z.-Y., Zhang, J., Wu, Y., Zhang, Y.-Y., Lin, J., and Liu, S.-M.: Hypoxia off the Changjiang (Yangtze River) Estuary: oxygen depletion and organic matter decomposition, Mar. Chem., 125, 108-116, doi:10.1016/j.marchem.2011.03.005, 2011.

---

## Author Response (AR2)

Jianzhong Su[1], Minhan Dai[1*], Biyan He[1, 2], Lifang Wang[1], Jianping Gan[3], Xianghui
Guo[1], Huade Zhao[1] and Fengling Yu[1]

[1]State Key Laboratory of Marine Environmental Science, Xiamen University, Xiamen,
China
[2]College of Food and Biological Engineering, Jimei University, Xiamen, China
[3]Department of Mathematics and Division of Environment, Hong Kong University of
Science and Technology, Kowloon, Hong Kong, China
*Correspondence to*: Minhan Dai (mdai@xmu.edu.cn)

Referee #1

Authors responded to all my original comments, but I feel that authors do not take into account main points of my comments.

- Uncertainties and difference between the CJE and PRE (similar issue was raised by reviewer #1): Authors re-estimated 13C value of organic carbon remineralized in the hypoxic zone to be -23.2 +/- 1.1, which was originally -21.8 +/- 0.6. Such a large change seems to be simply due to slight changes in end-member values. Authors may have to allow seasonal and spatial variations of end-members. Indeed, the large scattering and outlier evaluation in Figure 8 will result in much larger uncertainties in

13C value of OC remineralized. Authors should explain how the scattering and outlier are included in the uncertainty (+/- 1.1) of OC remineralized values calculated. In addition, I still do not see the visible difference between CJE and PRE in Figure 8

considering the scattering and natural variations. I think that authors try to obtain the accuracy that the method cannot provide.

[Response]: We have to clarify that we did not change the end-member values of our three end-member mixing model in either $\theta$, S, DIC or $\delta^{13}C_{DIC}$. These end-member values have been well justified by taking into account the uncertainties as explicitly explained in our revised submission.    The uncertainties of end-member values mostly come from averaging values of several stations at certain salinity, thus including spatial variations to some extent.

The reviewer might be right about the seasonality of the end-member values but here we are examining the summer bottom hypoxia where the mixing scheme of the water masses is rather clear as demonstrated in the T-S diagram shown in Fig. 7a.

A broader range of data at oxygen deficiency (n=38) (instead of those associated with severe oxygen deficiency at AOU> 100 µmol kg$^{-1}$ (n=9) in our original submission) is now being used to derive $\delta^{13}C_{OCx}$ considering on the main comments from Reviewer

#1. This re-estimation is justified because the oxygen depletion generating from organic matter remineralization distributes as a continuum in the water column and the accumulative oxygen reduction ultimately results in hypoxia formation.

As shown in Fig. 8, we plotted $\Delta$ ($\delta^{13}C_{DIC} \times DIC$) against $\Delta DIC$ including all the scattering points in the subsurface water in the PRE, and obtained a slope of -23.2±1.1 ‰

with a high correlation coefficient (R$^2$=0.86,    P< 0.001). Thus, the original $\delta^{13}C$

signature of the remineralized organic matter is -23.2 ‰ with an uncertainty of

±1.1 ‰.

The propagation error is shown in the following figure (Fig. S1). We are not sure how the reviewer concluded that there was no visible difference between PRE and CJE.

The linear regression slope of the PRE is -23.2±1.1 ‰, which is statistically different from that of the CJE (-18.5±1.0 ‰, R$^2$=0.6, P< 0.001). Our statistic has a confidence level of 95 %.

[Figure]

Figure S1. Propagation errors for $\Delta$ ($\delta^{13}C_{DIC} \times DIC$) and $\Delta DIC$ in Fig. 8.

- Use of Ca2+ as a conservative tracer: I am not convinced why authors use Ca, instead of salinity, as a conservative tracer although Ca is less conservative.

[Response]: We perfectly agreed that $Ca^{2+}$ is a less conservative tracer than salinity. The fact is that salinity has been applied as a conservative tracer in our three end-member mixing model, and thus we ought to find alternative independent tracers to validate the model. $Ca^{2+}$ is obviously such an alternative. As we explained in our previous responses, $Ca^{2+}$ is conservative in our system as supported by a strong linear relationship between surface water $Ca^{2+}$ and salinity, and aragonite oversaturation ($\Omega_{arag}$=2.6±0.7) in the subsurface water.

- Relative contribution of bottom sediment OM: Authors may compare the water column DO consumption rates relative to the bottom sediment DO consumption rates in order to show that the contribution of bottom sediments are negligible. Figure 4 shows stronger DO consumption in the bottom layer intuitively.

[Response]: In the present study, we used bottom water taken from Station A10 on 27 July and conducted on-deck incubation experiments to estimate total oxygen consumption rate, which was 9.8 $\mu$mol L$^{-1}$ d$^{-1}$. Such a water column oxygen consumption rate could well support the oxygen decline rate observed in situ at Station A10 in the hypoxic zone between 20 July and 27 July (Fig. 6), which was 7.7 $\mu$mol L$^{-1}$ d$^{-1}$. This first order comparison strongly suggests that water column oxygen consumption may be predominate in the formation of the hypoxia in the present case.

Note that we cannot distinguish between the contribution of resuspended sediments vs. sinking POC to oxygen depletion, because both of these isotopic signals would have been reflected in oxidation product (DIC) and the derived $\delta^{13}$C value of the remineralized OC ($\delta^{13}C_{OCx}$). Please see details in our previous response to comment (3) of Referee #2. Quantifying the relative contribution of bottom sediment OM is beyond the scope of our present study. Moreover, the bottom sediment DO consumption rate is still controversial due to its high spatial variability.

In responding to the concerns of the reviewer, we have added in the further revised manuscript our water column total oxygen consumption rate determined by on-deck incubation experiments using bottom water taken from Station A10 (Table S1). "As a first order comparison, the water column total oxygen consumption rate of 9.8 μmol $L^{-1}$ $d^{-1}$ could well support the oxygen decline rate observed at Station A10 in the hypoxic zone between 20 July and 27 July (Fig. 6), which was 7.7 μmol $L^{-1}$ $d^{-1}$. This comparison along with the stoichiometry between ΔDIC and AOU strongly suggests that water column aerobic respiration may be predominate in the formation of the hypoxia in the present case." Please see P4 L14-20 and P10 L12-18 in the further revised manuscript.

Table S1. Comparison of oxygen consumption rate between in situ observations and on-deck incubation experiments at Stations A10 in the hypoxic zone.

| Station | $DO_{ini}$ (μmol $L^{-1}$) | $DO_{end}$ (μmol $L^{-1}$) | Interval (d) | ΔDO (μmol $L^{-1}$) | DO decline rate (μmol $L^{-1}$ $d^{-1}$) | Total oxygen consumption rate (μmol $L^{-1}$ $d^{-1}$) |
|---------|---------|---------|----------|-----|-------------|-------------------|
| A10 | 156 | 101 | 7.1 | 55 | 7.7 | 9.8 |

Note that the subscripts "ini" means the initial sampling and "end" means the end sampling. The Interval shows the days between these two repeated sampling. DO decline rate equals to the amount of DO reduction dividing by interval.

- Significant figures: Authors show that DO and DIC precisions are +/- 1 and 2 respectively. But, they say that "we kept one decimal place for DO/DIC…". Authors should explain why they keep one decimal place although the values are meaningless.
[Response]: Accepted. Do not keep one decimal place of DO/DIC in the main text.

Referee #2
The revisions made in the paper by Su et al. have substantially improved the quality of the paper which, to my opinion, is now ready for publication. You have answered the main points raised in my previous review, especially the one dealing with end-members which is crucial in quantifying the uncertainty of the message provided by the paper. I can see that you have changed the value and expanded the uncertainty of your proportion of terrigeneous OM in the mix generating hypoxia.
I particularly appreciated the long and detailed response letter which made the revisions easy to check and evaluate.

I have just a few technical points which should further improve the clarity of your paper:

- You use very often "pre" for predicted (by the mixing model) and "PRE" for Pearl

River Estuary. It is confusing in the equations and Figures when endmembers are quoted (Eq. 7 page 10, or Figure caption and axis labels of Figure 7). In this Figure (7)

especially, "PRE" and "pre" co-exist in the axis title which should be changed. You should choose one of them (e.g. Pearl River Estuary) and replace "pre" by "mod" for model.

[Response]: Accepted. Use "con" (conservative) instead of "pre" (predicted)

throughout the manuscript.

- You use very often "PRE" for Pearl River Estuary and CJE for "ChangJiang Estuary".

At least for the Figure captions, I would recommend to replace CJE by ChangJiang

Estuary, as it is not so obvious for most readers and it would make the Figure easier to read.

[Response]: Accepted. Please see captions in Figs. 5, 7 and 8.

[revised manuscript text omitted]

---

## Author Response (AR3)

**Response to Editorial Comments:**

P7, L10: are complicated
[Response]: Done. Please see P2, L10.

P7, L26: A number of the phytoplankton-centric hypoxia
[Response]: Done. Please see P2, L26.

P8, L29: during 17–18 July
[Response]: Done. Please see P3, L29.

P8, L30: between 13–16 July
[Response]: Revised as between 13 and 16 July. Please see P3, L30.

P8, L31: between 19–27 July
[Response]: Revised as between 19 and 27 July. Please see P3, L31.

P9, L18: Please indicate the maximum differences in temperature between the bottom and surface waters, because temperature can significantly affect the microbial activity.
[Response]: Added as "Note that the maximum difference in temperature between the bottom and surface water was 3 ℃ during the incubation". Please see P4, L18-20.

P17, L21: than in the CJE (-8.8 ‰, -24.4±0.2 ‰) at S<0.4.
[Response]: Added the salinity range for the end-members in the CJE as "S<0.2"

P17, L27: temperatures
[Response]: Done. Please see P12, L27.

P17, L28 in the PRE were
[Response]: Done. Please see P12, L28.

P28: Insert a space between θ and (°C).
[Response]: Done. Please see P23.

P29: In references in Table 2, please use parentheses for the year published. For example, Hu et al. (2006).

[Response]: Done. Please see P24.

[Response]: Done. Please see P31, P32.

In addition, we determined not to specify the data points at AOU>100 µmol kg$^{-1}$ in Fig. 8, because the grey dots confused the comparison between those two systems.